# A New First-Order Meta-Learning Algorithm with Convergence Guarantees

**El Mahdi Chayti**                                             *el-mahdi.chayti@epfl.ch*

**Martin Jaggi**                                               *martin.jaggi@epfl.ch*

*Machine Learning and Optimization Laboratory (MLO), EPFL, Switzerland*

**Reviewed on OpenReview:** *https://openreview.net/forum?id=4zGAXQxisB*

## Abstract

Learning new tasks by leveraging prior experience is a fundamental trait of intelligent systems. While Model-Agnostic Meta-Learning (MAML) is a leading approach, it suffers from significant computational and memory overhead due to the requirement of computing second-order meta-gradients. We propose **FO-B-MAML**, a novel first-order variant of MAML derived from a bi-level optimization perspective. Our framework introduces a new expression of the meta-gradient, defined as the derivative of the solution of a perturbed optimization problem. This formulation allows the meta-gradient to be estimated using various finite difference methods; in this work, we propose and analyze two simple yet effective estimators: a forward and a symmetric approximation.

Unlike existing first-order methods like FO-MAML and Reptile, which suffer from irreducible bias, we prove that FO-B-MAML converges to a stationary point of the meta-objective. Notably, the symmetric estimator achieves an improved $\mathcal{O}(\delta^{2/3})$ bias rate, strictly enhancing previous first-order theory. Furthermore, we demonstrate that the MAML objective violates standard smoothness assumptions; we show instead that its smoothness constant grows with the norm of the meta-gradient. This property theoretically justifies the use of normalized or clipped-gradient methods (SNGDM) over vanilla gradient descent.

Our empirical results validate these advancements: FO-B-MAML achieves high accuracy, closely following second-order MAML performance. Crucially, our method bypasses the "activation bottleneck" of second-order approaches, maintaining a flat memory footprint even when scaling to deep, activation-heavy CNNs and Transformers.

## 1 Introduction

A hallmark of intelligence is the ability to swiftly acquire new skills by leveraging prior experiences from similar tasks. Recent research has explored how meta-learning algorithms (Schmidhuber, 1987; Thrun & Pratt, 1998; Naik & Mammone, 1992) can replicate this capability by learning to learn across a variety of tasks efficiently. This mastery allows for learning a novel task with only minimal training data, sometimes just a single example, as demonstrated in (Santoro et al., 2016; Vinyals et al., 2016; Finn et al., 2017b).

Meta-learning approaches can be generally categorized into three primary frameworks:

- **Metric-learning approaches**: These methods learn a structured embedding space where non-parametric techniques, such as nearest neighbors, perform effectively (Koch, 2015; Vinyals et al., 2016; Snell et al., 2017; Oreshkin et al., 2018; Allen et al., 2019).

- **Black-box approaches**: These involve training recurrent or recursive neural networks to either produce weight updates directly from input data (Hochreiter et al., 2001; Andrychowicz et al., 2016; Li & Malik, 2016; Ravi & Larochelle, 2017) or to generate predictions for new inputs (Santoro et al., 2016; Duan et al., 2016; Wang et al.,

2016; Munkhdalai & Yu, 2017; Mishra et al., 2017). Attention-based models (Vinyals et al., 2016; Mishra et al., 2017) are also frequently employed in this context.

- **Optimization-based approaches**: These methods utilize bi-level optimization to integrate learning procedures, such as gradient descent, into the meta-optimization problem (Finn et al., 2017b; Finn, 2018; Bertinetto et al., 2018; Zintgraf et al., 2018; Li et al., 2017; Finn et al., 2018; Zhou et al., 2018; Harrison et al., 2018). The "inner" optimization handles task adaptation, while the "outer" objective is the meta-training goal: the average test loss after adaptation over a set of tasks. This approach, exemplified by (Maclaurin et al., 2015) and MAML (Finn et al., 2017b), learns optimal initial model parameters to facilitate rapid adaptation and generalization.

Beyond these categories, hybrid strategies have been explored to combine the strengths of different methods (Rusu et al., 2018; Triantafillou et al., 2019). In this study, we concentrate on **optimization-based methods**, specifically MAML (Finn et al., 2017b), which has been demonstrated to possess the expressive power of black-box strategies (Finn & Levine, 2017). Additionally, MAML is versatile across various scenarios (Finn et al., 2017c; Mi et al., 2019; Al-Shedivat et al., 2017; Finn et al., 2019) and ensures a consistent optimization process (Finn, 2018).

While meta-learning initialization holds promise, it necessitates backpropagation through the inner optimization algorithm, introducing significant challenges. This includes the requirement for higher-order derivatives, leading to substantial computational and memory overheads and potential issues like vanishing gradients. Consequently, scaling optimization-based meta-learning to tasks with large datasets or numerous inner-loop steps becomes arduous.

These challenges can be partially mitigated by taking few gradient steps (Finn et al., 2017b), truncating the backpropagation process (Shaban et al., 2018), using implicit gradients (Rajeswaran et al., 2019a), or omitting higher-order derivative terms (Finn et al., 2017b; Nichol et al., 2018). However, these approximations may lead to sub-optimal performance (Wu et al., 2018). An additional challenge relates to the convergence analysis; while it was shown in (Fallah et al., 2019) that the MAML objective can be non-smooth even with one step, other works such as (Rajeswaran et al., 2019a) simply assume smoothness for multi-step trajectories. A comprehensive convergence analysis remains missing.

To address these limitations, we design a new first-order MAML algorithm that is path-independent and provides rigorous theoretical convergence guarantees.

**Contributions:**

1. **Perturbation-Based Meta-Gradient Identity:** We derive a novel expression for the MAML meta-gradient (Proposition 1) by reformulating it as the derivative of a perturbed inner-optimization solution. This identity decouples the meta-gradient from the specific inner-loop optimization trajectory, providing a path-independent framework for meta-learning.

2. **Memory-Efficient First-Order Estimators:** We introduce two first-order MAML variants—Forward and Symmetric—that estimate the meta-gradient using finite differences between perturbed point-estimates. Unlike heuristic shortcuts such as FO-MAML or Reptile (Nichol et al., 2018), our symmetric estimator offers a controllable bias that approaches the true meta-gradient as the inner solver improves.

3. **Theoretical Foundation for Generalized Smoothness:** We establish that the MAML objective satisfies a *generalized smoothness* condition rather than standard Lipschitz smoothness. While prior work (Fallah et al., 2019) identified non-smoothness primarily in the single-step GD case, our analysis provides a formal justification for why meta-gradient clipping stabilizes MAML in general deep learning settings, linking our findings to established non-standard optimization theory (Zhang et al., 2020b).

4. **Superior Convergence Rates:** We provide provable convergence rates for FO-B-MAML to stationary points. While standard analysis for first-order MAML variants typically yields a complexity of $O(\varepsilon^{-6})$ (Fallah et al., 2019), our forward estimator matches this baseline, and our symmetric estimator improves the complexity to $O(\varepsilon^{-5.5})$. This demonstrates that the symmetric estimator is not only more accurate in its gradient approximation but also theoretically faster than existing first-order counterparts.

5. **Experimental Validation of Scalability:** We validate experimentally that our algorithm scales significantly better than second-order variants in terms of memory usage while maintaining comparable performance.

## 2 Related Work

The landscape of optimization-based meta-learning has evolved primarily around addressing the computational and memory bottlenecks of second-order methods while maintaining gradient accuracy.

**MAML and First-Order Heuristics.** The seminal Model-Agnostic Meta-Learning (MAML) framework (Finn et al., 2017a) relies on differentiating through the entire inner-loop optimization trajectory. While effective, the requirement to compute and store higher-order derivatives leads to a memory footprint that scales poorly with inner-loop steps and model complexity. First-order heuristics such as FO-MAML (Finn et al., 2017a) and Reptile (Nichol et al., 2018) mitigate these costs by dropping higher-order terms or using the distance between the initial and adapted parameters as a gradient direction. However, these methods often suffer from heuristic bias and lack a formal optimization foundation.

**Implicit and Perturbation-Based Bilevel Optimization.** Our work is situated within the broader context of bilevel programming for hyperparameter optimization and meta-learning (Franceschi et al., 2018). To avoid the path-dependence of unrolled optimization, implicit MAML (iMAML) (Rajeswaran et al., 2019b) utilizes the Implicit Function Theorem (IFT) to define meta-gradients based on the final adapted solution. While iMAML avoids storing the inner trajectory, it requires solving an expensive linear system, often via Conjugate Gradient (CG) or Neumann-series approximations, which can be computationally intensive and sensitive to damping parameters (Lorraine et al., 2020).

A closely related line of work explores hypergradient estimation through perturbation analysis. Specifically, Kwon et al. (2023) introduced a framework for stochastic bilevel optimization that interprets the hypergradient as the sensitivity of the lower-level solution to perturbations. Our method, FO-B-MAML, shares this perturbation perspective but introduces several key distinctions. First, while Kwon et al. (2023) derive their theory specifically for Gradient Descent (GD) as the inner solver, our bi-level formulation treats the inner task as a general optimization problem, making the derivation agnostic to the specific optimization trajectory. Second, we propose and analyze a *symmetric-difference estimator*, which we prove offers a superior bias-rate ($O(\delta^{2/3})$ vs $O(\delta^{1/2})$) when the inner problem is solved approximately—a common necessity in large-scale deep learning.

**Memory Scalability and Modern Architectures.** Recent efforts have targeted the "activation bottleneck" in meta-learning, where storing activations for second-order backpropagation becomes the primary constraint (Finn et al., 2017a). This is particularly acute in Transformers, where self-attention incurs a quadratic memory cost $O(L^2)$ relative to sequence length $L$ (Vaswani et al., 2017). FO-B-MAML bypasses this by requiring only two parameter point-estimates rather than storing the full inner-loop computation graph. This enables meta-learning on high-dimensional, activation-heavy architectures where standard MAML would trigger Out-of-Memory (OOM) errors as sequence length scales.

**Smoothness and Optimization Stability.** Finally, our work contributes to the theoretical understanding of the MAML landscape. While standard bilevel analysis often assumes global Lipschitz smoothness, we establish that the meta-learning objective exhibits *generalized smoothness*—where the smoothness constant scales with the gradient norm. This finding aligns with the generalized smoothness framework of Zhang et al. (2020b) and provides a rigorous theoretical justification for the use of clipped or normalized gradient methods to stabilize meta-training.

## 3 Preliminaries

### 3.1 Vanilla MAML

We assume we have a set of training tasks $\{\mathscr{T}_i\}_{i=1}^M$ drawn from an unknown distribution of tasks $P(\mathscr{T})$, such that for each task $\mathscr{T}_i$ one can associate a training $\mathscr{D}_i^{\text{tr}}$ and test $\mathscr{D}_i^{\text{test}}$ dataset—or equivalently a training $\hat{f}_i$ and test $f_i$ objective. Then, the vanilla MAML objective is that of solving the following optimization problem:

$$\boldsymbol{\theta}^\star := \underset{\boldsymbol{\theta} \in \Theta}{\arg\min}\, F(\boldsymbol{\theta})$$

$$F(\boldsymbol{\theta}) := \frac{1}{M} \sum_{i=1}^M \left[ F_i(\boldsymbol{\theta}) := f_i\left( \phi_i(\boldsymbol{\theta}) = \mathscr{A}lg(\hat{f}_i, \boldsymbol{\theta}, \boldsymbol{h}) \right) \right], \tag{1}$$

where $\mathscr{A}lg(\hat{f}_i, \boldsymbol{\theta}, \boldsymbol{h})$ is an optimization algorithm that takes as input the objective $\hat{f}_i$ (i.e., dataset and loss), the initialization $\boldsymbol{\theta}$ and other hyperparameters denoted by $\boldsymbol{h}$ (.e.g., learning rate, and the number of steps), then outputs an

updated task-specific parameter $\phi_i(\boldsymbol{\theta})$ that is hopefully a better "approximate" solution for the individual task objective $\hat{f}_i$.

For example, $\mathscr{A}lg(\hat{f}_i, \boldsymbol{\theta}, \cdot)$ may correspond to one or multiple steps of gradient descent on $\hat{f}_i$ initialized at $\boldsymbol{\theta}$. For example, if we use one step of gradient descent with a learning rate $\alpha$, then we have:

$$\phi_i(\boldsymbol{\theta}) \equiv \mathscr{A}lg(\hat{f}_i, \boldsymbol{\theta}, \boldsymbol{h} := \{\alpha\}) = \boldsymbol{\theta} - \alpha \nabla_{\boldsymbol{\theta}} \hat{f}_i(\boldsymbol{\theta}). \tag{2}$$

To solve (1) with gradient-based methods, we require a way to differentiate through $\mathscr{A}lg$. In the case of multiple steps like (2), this corresponds to backpropagating through the dynamics of gradient descent. This backpropagation through gradient-based optimization algorithms naturally involves higher order derivatives and the need to save the whole trajectory to compute the meta-gradient (i.e., the gradient of $F$), which is a big drawback to vanilla MAML.

Another (potential) drawback of MAML defined in (1) is that it depends on the choice of the optimization algorithm $\mathscr{A}lg$ (since we need its specific trajectory).

## 3.2   First-order MAML and Reptile

One option considered in the literature to address the computational and memory overheads encountered when differentiating through gradient-based optimization algorithms is to devise first-order meta-gradient approximations (Nichol et al., 2018). Two such approaches stand out: **FO-MAML** and **Reptile**.

**FO-MAML** simply ignores the Jacobian $\dfrac{d\mathscr{A}lg}{d\boldsymbol{\theta}}$ leading to the following approximation:

$$\boldsymbol{g}_{\text{FO-MAML}} = \frac{1}{M} \sum_{i=1}^{M} \nabla f_i(\phi_i(\boldsymbol{\theta})) \tag{3}$$

The **Reptile** approximation is less straightforward, but the crux of it is using an average gradient over the inner optimization algorithm's trajectory.

$$\boldsymbol{g}_{Reptile} = \frac{1}{M} \sum_{i=1}^{M} \frac{\boldsymbol{\theta} - \phi_i(\boldsymbol{\theta})}{K\alpha} \tag{4}$$

where in (4), $K$ is the number of steps of $\mathscr{A}lg$ and $\alpha$ is the learning rate.

These two approximations avoid the prohibitive computational and memory costs associated with vanilla MAML. However, both approximations introduce bias to the true meta-gradient that is irreducible, at least in the case of FOMAML (Fallah et al., 2019) (the bias of Reptile is not clear in general).

## 3.3   B-MAML: MAML as a fully Bi-Level Optimization Problem

Ignoring the computational overhead of MAML, the memory overhead is naturally a result of the dependence of the MAML objective in (1) on the choice of the inner optimization algorithm $\mathscr{A}lg$ and thus on the trajectory of $\mathscr{A}lg$; then if one can break this dependence on $\mathscr{A}lg$, one would avoid this memory overhead; one idea to accomplish the latter is to make the MAML objective depend on the inner optimization problem rather than the specific optimization algorithm used to solve such an optimization problem. This will amount to framing MAML as the following purely bi-level optimization problem : (outer-level)

$$\boldsymbol{\theta}^{\star} := \arg\min_{\boldsymbol{\theta} \in \Theta} F(\boldsymbol{\theta}) := \frac{1}{M} \sum_{i=1}^{M} \left[ F_i(\boldsymbol{\theta}) := f_i\left( \phi_i^{\star}(\boldsymbol{\theta}) \right) \right] \tag{5}$$

where for $i \in \{1, \cdots, M\}$ we define (recall that $f$ and $\hat{f}$ denote validation and training objectives respectively):

$$\phi_i^{\star}(\boldsymbol{\theta}) := \arg\min_{\phi \in \Theta} \hat{f}_i(\phi) + \frac{\lambda}{2} \|\phi - \boldsymbol{\theta}\|^2 \quad \text{(inner-level)} \tag{6}$$

Where $\lambda$ is a real hyperparameter that plays the role of the inverse of the learning rate $\alpha$ that MAML had, it helps control the strength of the meta-parameter or prior ($\boldsymbol{\theta}$) relative to new data. We note that this hyperparameter can be a vector or a matrix, but we keep it as a scalar for simplicity.

We also note that the formulation (6) is not new and was introduced, for example, in (Rajeswaran et al., 2019a).

To use gradient-based methods to solve (5), we need to compute the gradient $\nabla F$. Using the implicit function theorem and assuming that $\lambda I + \nabla^2 \hat{f}_i$ is invertible, it is easy to show that

$$\nabla F_i(\boldsymbol{\theta}) = \left( I + \frac{1}{\lambda} \nabla^2 \hat{f}_i(\boldsymbol{\phi}_i^\star(\boldsymbol{\theta})) \right)^{-1} \nabla f_i(\boldsymbol{\phi}_i^\star(\boldsymbol{\theta})) \tag{7}$$

What is noteworthy about (7) is that the meta-gradient $\nabla F_i(\boldsymbol{\theta})$ only depends on $\boldsymbol{\phi}_i^\star(\boldsymbol{\theta})$ which can be estimated using any optimization algorithm irrespective of the trajectory said-algorithm will take.

A downside of (7) is that it involves computing the Hessian and inverting it. To reduce this burden, one can equivalently treat $\nabla F_i(\boldsymbol{\theta})$ as a solution to the following linear system (with the unknown $\boldsymbol{v} \in \mathbb{R}^d$):

$$\left( I + \frac{1}{\lambda} \nabla^2 \hat{f}_i(\boldsymbol{\phi}_i^\star(\boldsymbol{\theta})) \right) \boldsymbol{v} = \nabla f_i(\boldsymbol{\phi}_i^\star(\boldsymbol{\theta})), \tag{8}$$

which only needs access to Hessian-Vector products instead of the full Hessian and can be approximately solved with the conjugate gradient algorithm, for example, which is exactly the idea of (Rajeswaran et al., 2019a).

In this work, we propose a different strategy that consists of writing the meta-gradient as the gradient of the solution of a perturbed optimization problem with respect to its perturbation parameter. This means we can approximate the meta-gradient using the solution of two optimization problems.

### 3.4 Comparison to ES-MAML

The closest method in the literature to ours is perhaps ES-MAML(Song et al., 2020), which is based on the idea of *evolution strategies* that consists of replacing a given function $h$ and with its Gaussian smoothing: $h_v(\boldsymbol{\theta}) := \mathbb{E}_{\boldsymbol{g} \sim \mathcal{N}(\boldsymbol{0}, \boldsymbol{I})}[h(\boldsymbol{\theta} + v\boldsymbol{g})]$. which has the property of being smooth (even when $h$ is not) with: $\nabla h_v(\boldsymbol{\theta}) := \frac{1}{v} \mathbb{E}_{\boldsymbol{g} \sim \mathcal{N}(\boldsymbol{0}, \boldsymbol{I})}[h(\boldsymbol{\theta} + v\boldsymbol{g})\boldsymbol{g}]$. In our case $h(\boldsymbol{\theta}) := F(\boldsymbol{\theta}) := \frac{1}{m} \sum_{i=1}^m f_i(\boldsymbol{\phi}_i^\star(\boldsymbol{\theta}))$. Thus ES-MAML minimizes the following perturbed problem: $F_v(\boldsymbol{\theta}) = \frac{1}{m} \sum_{i=1}^m \mathbb{E}_{\boldsymbol{g} \sim \mathcal{N}(\boldsymbol{0}, \boldsymbol{I})}[f_i(\boldsymbol{\phi}_i^\star(\boldsymbol{\theta} + v\boldsymbol{g}))]$.

Noting that

$$\boldsymbol{\phi}_i^\star(\boldsymbol{\theta} + v\boldsymbol{g}) = \arg\min_{\boldsymbol{\phi}} -v\lambda \langle \boldsymbol{g}, \boldsymbol{\phi} \rangle + \hat{f}_i(\boldsymbol{\phi}) + \frac{\lambda}{2} \|\boldsymbol{\phi} - \boldsymbol{\theta}\|^2. \tag{9}$$

we see that ES-MAML can be seen indeed as using our idea of perturbing the inner objective using the (random) function $\boldsymbol{\phi} \mapsto -\lambda \langle \boldsymbol{g}, \boldsymbol{\phi} \rangle$ and the perturbation parameter being $v$. However, in our case, we will perturb the inner objective using the outer objective directly, this will result in a better algorithm that needs to only call the inner solver twice instead of many times (to estimate the average over the Gaussian distribution).

ES-MAML(Song et al., 2020) is mainly used in reinforcement learning where it is observed to improve exploration but has not been considered in a broader sense as it potentially needs many random Gaussian samples and solving the inner problem and evaluating the test function as many times (note that for neural networks evaluating a function and computing its gradients have approximately the same complexity). We show that ES-MAML fails to perform in our simple synthetic experiment in Figure1.

## 4 First-Order B-MAML

We consider the B-MAML objective defined in (5), (6); our main idea relies on perturbing the inner optimization problem (6). For a perturbation parameter $v \in \mathbb{R}$, and each training task $\mathcal{T}_i$, we introduce the following perturbed inner optimization problem, which interpolates between validation and training objectives:

$$\boldsymbol{\phi}_{i,v}^\star(\boldsymbol{\theta}) := \arg\min_{\boldsymbol{\phi} \in \Theta} v f_i(\boldsymbol{\phi}) + \hat{f}_i(\boldsymbol{\phi}) + \frac{\lambda}{2} \|\boldsymbol{\phi} - \boldsymbol{\theta}\|^2. \tag{10}$$

To provide an intuition about this choice of perturbation, let's assume the test loss $f_i$ is linear i.e.: $\nabla f_i(\boldsymbol{\phi}) = \boldsymbol{g}_f$ a constant, then it is easy to show that:

$$\boldsymbol{\phi}_{i,v}^\star(\boldsymbol{\theta}) = \boldsymbol{\phi}_i^\star(\boldsymbol{\theta} - v\frac{\boldsymbol{g}_f}{\lambda}) \tag{11}$$

Which is the perturbed problem (9) with $g = -\frac{g_f}{\lambda}$. This means that instead of picking a random perturbation direction like ES-MAML, we pick an informed and specific direction that we will show leads to a better estimate of the gradient.

We assume that there is a neighbourhood $\mathscr{V}_0$ of 0, such that $\phi^\star_{i,v}(\boldsymbol{\theta})$ is well-defined for any $v \in \mathscr{V}_0$ and $\boldsymbol{\theta} \in \Theta$. Then we can show the following result:

> **Proposition 1** *For any training task $\mathscr{T}_i$, if $\nabla F_i(\boldsymbol{\theta})$ exists, then $v \mapsto \phi^\star_{i,v}(\boldsymbol{\theta})$ is differentiable at $v = 0$ and $\nabla F_i(\boldsymbol{\theta}) = -\lambda \left.\frac{d\phi^\star_{i,v}(\boldsymbol{\theta})}{dv}\right|_{v=0}$.*

**Sketch of the proof.** We use the fact that $\phi^\star_{i,v}(\boldsymbol{\theta})$ is a stationary point of $\phi \mapsto v f_i(\phi) + \hat{f}_i(\phi) + \frac{\lambda}{2}\|\phi - \boldsymbol{\theta}\|^2$ this will give us a quantity that is null for all $v \in \mathscr{V}_0$, then we differentiate with respect to $v$ and set $v = 0$.

Proposition 1 writes the meta-gradient $\nabla F_i(\boldsymbol{\theta})$ as the derivative of another function that is a solution to the perturbed optimization problem 10, thus presenting us with a way we can approximate the meta-gradient using the finite difference method (Atkinson & Han, 2001). We consider mainly two approximations: the **forward** and **symmetric** approximations successively defined as follows:

$$g^{\text{For}}_{i,v}(\boldsymbol{\theta}) = -\lambda \left(\frac{\phi^\star_{i,v}(\boldsymbol{\theta}) - \phi^\star_{i,0}(\boldsymbol{\theta})}{v}\right) \tag{12}$$

$$g^{\text{Sym}}_{i,v}(\boldsymbol{\theta}) = -\lambda \left(\frac{\phi^\star_{i,v}(\boldsymbol{\theta}) - \phi^\star_{i,-v}(\boldsymbol{\theta})}{2v}\right) \tag{13}$$

We note that more involved approximations (that need solving more than two optimization problems) can be engineered, but we limit ourselves, in this work, to (12) and (13).

Assuming that $v \mapsto \phi^\star_{i,v}(\boldsymbol{\theta})$ is regular enough near $v = 0$ (for example, three times differentiable on $\mathscr{V}_0$ and its third derivative is bounded), then we should expect that

$$\|\nabla F_i(\boldsymbol{\theta}) - g^{\text{For}}_{i,v}(\boldsymbol{\theta})\| = \mathscr{O}(\lambda v) \quad \text{and} \quad \|\nabla F_i(\boldsymbol{\theta}) - g^{\text{Sym}}_{i,v}(\boldsymbol{\theta})\| = \mathscr{O}(\lambda v^2) \tag{14}$$

We will provide conditions under which we get the first identity in Equation 14 in Section 5 and support the second experimentally in Section 6.

In practice, we can't realistically solve the optimization problem (10) and have access to the true values of $\phi^\star_{i,v}(\boldsymbol{\theta})$; instead, we will solve the problem (10) approximatively using any algorithm $\mathscr{A}lg$ of our choice and assume that for a given precision $\delta$, we can get an approximate solution $\tilde{\phi}_{i,v}(\boldsymbol{\theta})$ of (10) such that

$$\|\tilde{\phi}_{i,v}(\boldsymbol{\theta}) - \phi^\star_{i,v}(\boldsymbol{\theta})\| \le \delta . \tag{15}$$

**Note.** In the stochastic case, we need to replace the guarantee in (15) by $\mathbb{E}\|\tilde{\phi}_{i,v}(\boldsymbol{\theta}) - \phi^\star_{i,v}(\boldsymbol{\theta})\| \le \delta$ where $\mathbb{E}$ is the expectation over the randomness of the algorithm $\mathscr{A}lg$; most importantly, this will not change anything in our analysis except for the additional expectations.

We use $\tilde{g}^{\text{method}}_{i,v}$ for method $\in \{\text{For}, \text{Sym}\}$, to denote the estimator resulting from the use of the approximate solutions; this will introduce an additional bias (w/t $\nabla F_i$) to our estimators in (12), (13); it is easy to show that this bias should be bounded by $\frac{2\lambda\delta}{v}$ in the worst case. Notice that this bias term increases with small values of $v$, which suggests a sweet spot for $v$ when including the bias terms in (14).

The overall bias is then

$$\|\nabla F_i(\boldsymbol{\theta}) - \tilde{g}^{\text{For}}_{i,v}(\boldsymbol{\theta})\| = \mathscr{O}\left(\lambda v + \frac{\lambda\delta}{v}\right) \quad \text{and} \quad \|\nabla F_i(\boldsymbol{\theta}) - \tilde{g}^{\text{Sym}}_{i,v}(\boldsymbol{\theta})\| = \mathscr{O}\left(\lambda v^2 + \frac{\lambda\delta}{v}\right), \tag{16}$$

minimizing for $v$ we get that for $v^{For} \sim \sqrt{\delta}$ and $v^{Sys} \sim \delta^{1/3}$ we get:

$$\|\nabla F_i(\boldsymbol{\theta}) - \tilde{g}^{\text{For}}_{i,v}(\boldsymbol{\theta})\| = \mathscr{O}(\lambda\sqrt{\delta}) \quad \text{and} \quad \|\nabla F_i(\boldsymbol{\theta}) - \tilde{g}^{\text{Sym}}_{i,v}(\boldsymbol{\theta})\| = \mathscr{O}(\lambda\delta^{2/3}), \tag{17}$$

---

**Algorithm 1** First Order Bi-Level MAML (FO-B-MAML)

---

1: **Require:** Distribution over tasks $P(\mathcal{T})$, outer step size $\eta$, regularization strength $\lambda$,
2: **Hyperparameters:** Precision $\delta$ and small perturbation parameter $\nu$
3: **while** not converged **do**
4:     Sample mini-batch of tasks $\{\mathcal{T}_i\}_{i=1}^B \sim P(\mathcal{T})$
5:     **for** Each task $\mathcal{T}_i$ **do**
6:         Use an iterative solver to get $\tilde{\phi}_{i,\nu}(\boldsymbol{\theta})$ and $\tilde{\phi}_{i,0}(\boldsymbol{\theta})$ (or $\tilde{\phi}_{i,-\nu}(\boldsymbol{\theta})$) satisfying (15)
7:         Set $g_i = -\lambda(\tilde{\phi}_{i,\nu}(\boldsymbol{\theta}) - \tilde{\phi}_{i,0}(\boldsymbol{\theta}))/\nu$ $\left(\text{or } g_i = -\lambda(\tilde{\phi}_{i,\nu}(\boldsymbol{\theta}) - \tilde{\phi}_{i,-\nu}(\boldsymbol{\theta}))/(2\nu)\right)$
8:     **end for**
9:     Average above gradients to get $\hat{\nabla}F(\boldsymbol{\theta}) = (1/B)\sum_{i=1}^B g_i$
10:     Update meta-parameters with a gradient-based optimization algorithm of choice like GD, ClippedGD, or Adam.
11: **end while**

---

We summarize this in Algorithm 1.

In line 10 of Algorithm 1, we can use any optimization algorithm to update the meta-parameters $\boldsymbol{\theta}$, but our theoretical analysis in Section 5 will only consider the Gradient Descent (GD) and Clipped Gradient (ClippedGD) algorithms (or Normalized GD which is equivalent). We will show that the B-MAML objective (5) has a smoothness parameter that grows with the norm of its gradient; under this type of smoothness, it is known that ClippedGD is well-suited (Jingzhao et al., 2020), Adam (Kingma & Ba, 2015) is also well-suited since it estimates the curvature and uses it to normalize the gradient. We also show that under stronger assumptions, the B-MAML objective is smooth in the classical sense (meaning its gradient is Lipschitz), and in this case, GD can be used but can have a worse complexity.

Now that we have presented our main algorithm, it is time to discuss its theoretical guarantees under common assumptions used in the Bi-Level optimization literature.

## 5 Theoretical Analysis

In this Section, we provide theoretical guarantees of Algorithm 1 using both the forward approximation in equation (12) and the symmetric approximation in equation (13). We start by stating the assumptions that we make on the training tasks $\{\mathcal{T}_i\}_{i=1}^M$, then discuss the smoothness properties of the B-MAML objective defined in (6) resulting from such assumptions. Finally, we discuss the convergence rate when using Gradient Descent (GD) or Clipped Gradient Descent (SNGDM) as the meta-optimizer.

### 5.1 Assumptions

We will make use of the following assumptions:

> **Assumption 1 (Behavedness of train sets for Forward Case)** *For all training tasks $\mathcal{T}_i$, the training objective $\hat{f}_i$ is twice differentiable, $\hat{L}_1$-smooth and has $\hat{L}_2$-Lipschitz Hessian.*

> **Assumption 2 (Higher-Order Regularity for Symmetric Case)** *For the symmetric estimator to achieve its improved rate, we further assume for all training tasks $\mathcal{T}_i$ that the training objective $\hat{f}_i$ is $C^4$ with bounded fourth derivative (i.e., third derivative is $\hat{L}_3$-Lipschitz).*

> **Assumption 3 (Behavedness of test sets)** *For all training tasks $\mathcal{T}_i$, the test objective $f_i$ is differentiable, $L_0$–Lipschitz, and has $L_1$–smooth Hessian. For the symmetric estimator, we additionally assume $f_i$ is $C^3$ with a Lipschitz second derivative ($L_2$).*

> **Assumption 4 (Strong convexity)** *There exists $\mu > 0$ such that for all training tasks $\mathcal{T}_i$, the inner training objective $\hat{f}_i + \frac{\lambda}{2}\|\cdot\|^2$ is $\mu$-strongly convex.*

Under Assumption 1, taking $\lambda > \hat{L}_1$ ensures Assumption 4 holds with $\mu = \lambda - \hat{L}_1$. This guarantees that the perturbed problem (10) has a unique solution path $\nu \mapsto \phi_{i,\nu}^\star(\boldsymbol{\theta})$ that is $C^3$ for $\nu \in \mathcal{V}_0 = (-\frac{\mu}{L_1}, \frac{\mu}{L_1})$.

**Assumption 5 (Task similarity)** *There exists $\zeta \geq 0$, such that for all $\boldsymbol{\theta} \in \Theta$, we have:* $\frac{1}{M} \sum_{i=1}^{M} \|\nabla F_i(\boldsymbol{\theta}) - \nabla F(\boldsymbol{\theta})\|^2 \leq \zeta^2$.

**Assumption 6 (Bounded variance)** *We also assume that we have access to stochastic gradients of $f_i, \hat{f}_i$ with a variance bounded by $\sigma^2$.*

**Discussion of Assumptions.** The assumptions outlined above ensure the mathematical well-definedness of the bi-level objective and provide a rigorous foundation for our convergence analysis. Assumptions 1, 2, 3, and 6 are standard regularity conditions in non-convex and bi-level optimization literature, ensuring stable curvature and bounded variance for stochastic updates. Assumption 4 (Strong Convexity) is logically required to ensure a unique inner solution mapping $\boldsymbol{\theta} \mapsto \boldsymbol{\phi}^\star$, a prerequisite for a well-defined meta-gradient. While deep networks are non-convex, the proximal term $\frac{\lambda}{2}\|\boldsymbol{\phi} - \boldsymbol{\theta}\|^2$ effectively "convexifies" the local landscape, matching the theoretical requirements utilized in implicit differentiation frameworks like *iMAML*Rajeswaran et al. (2019a). Assumption 5 (Task Similarity) is borrowed from the Federated Learning literature to bound task heterogeneity. This allows us to formally control the variance introduced during the meta-gradient averaging step.

## 5.2 Properties of B-MAML and Estimator Bias

We characterize the bias for both the forward and symmetric estimators by bounding the derivatives of the solution path $z(v) = \boldsymbol{\phi}^\star_{i,v}(\boldsymbol{\theta})$ against the true meta-gradient $\nabla F_i(\boldsymbol{\theta}) = -\lambda z'(0)$.

**Proposition 2 (Refined Bias Expressions)** *Under the regularity assumptions, let $\mu$ be the strong convexity constant of the inner objective. The bias of the estimators relative to the true meta-gradient $\nabla F_i(\boldsymbol{\theta})$ satisfies:*

- *Forward Estimator:*

$$\|\nabla F_i(\boldsymbol{\theta}) - \boldsymbol{g}^{For}_{i,v}(\boldsymbol{\theta})\| = \mathcal{O}\left( \lambda v \left[ \frac{L_1}{\mu} + \frac{L_0 \hat{L}_2}{\mu^2} \right] \right) \tag{18}$$

- *Symmetric Estimator:*

$$\|\nabla F_i(\boldsymbol{\theta}) - \boldsymbol{g}^{Sym}_{i,v}(\boldsymbol{\theta})\| = \mathcal{O}\left( \lambda v^2 \left[ \frac{L_2}{\mu} + \frac{L_1 \hat{L}_2 + L_0 \hat{L}_3}{\mu^2} + \frac{L_0 \hat{L}_2^2}{\mu^3} \right] \right) \tag{19}$$

Balancing these with the numerical error $\mathcal{O}(\lambda \delta / v)$ from the solver precision $\delta$ yields the following:

**Corollary 1 (Total Bias of FO-B-MAML)** *With inner precision $\delta$ and optimal $v$:*

- *Forward Case:* $v \sim \delta^{1/2} \implies \|\nabla F_i(\boldsymbol{\theta}) - \tilde{g}^{For}_{i,v}(\boldsymbol{\theta})\| = \mathcal{O}(\lambda \delta^{1/2})$.

- *Symmetric Case:* $v \sim \delta^{1/3} \implies \|\nabla F_i(\boldsymbol{\theta}) - \tilde{g}^{Sym}_{i,v}(\boldsymbol{\theta})\| = \mathcal{O}(\lambda \delta^{2/3})$.

**Discussion.** The symmetric approximation provides a significant theoretical improvement, reaching an $\mathcal{O}(\delta^{2/3})$ bias rate. These results on the accelerated symmetric approximation are new and strictly improve over previous MAML theory, allowing for higher precision with first-order memory costs.

We now discuss the smoothness of the meta-objective. Proposition 3 shows that curvature scales with the gradient norm.

**Proposition 3** *Under Assumptions 1 and 3, for $\lambda \geq 2\hat{L}_1$, for any training task $\mathcal{T}_i$:*

$$\|\nabla F_i(\boldsymbol{\theta}) - \nabla F_i(\boldsymbol{\theta}')\| \leq \min(\mathscr{L}(\boldsymbol{\theta}), \mathscr{L}(\boldsymbol{\theta}'))\|\boldsymbol{\theta} - \boldsymbol{\theta}'\|,$$

where $\mathscr{L}(\boldsymbol{\theta}) = L_1/4 + \frac{\hat{L}_2}{4\lambda}\|\nabla F_i(\boldsymbol{\theta})\| := \mathscr{L}_0 + \mathscr{L}_1\|\nabla F_i(\boldsymbol{\theta})\|$. *Under Assumption 4, the meta-gradient is bounded by* $G = \frac{\lambda L_0}{\mu}$, *implying classical smoothness* $\|\nabla F(\boldsymbol{\theta}) - \nabla F(\boldsymbol{\theta}')\| \leq \mathscr{L}\|\boldsymbol{\theta} - \boldsymbol{\theta}'\|$ *with* $\mathscr{L} = \mathscr{L}_0 + G\mathscr{L}_1$.

## 5.3 Convergence Results

Based on the characterization of the $(\mathscr{L}_0, \mathscr{L}_1)$-smoothness and the estimator bias, we reduce the bi-level optimization problem in (5) to a single-level non-convex optimization problem. Throughout this section, we denote $\Delta = F(\boldsymbol{\theta}_0) - F^\star$, where $F^\star$ is the global minimum. We denote $\tilde{\sigma}^2 := \frac{M-B}{MB}\zeta^2$ as the outer variance from task sampling and recall that Assumption 6 implies an inner stochastic variance of $\mathscr{O}(\sigma^2)$.

---

**Theorem 1 (FO-B-MAML using ClippedGD)** *Under the generalized smoothness (Proposition 3), Algorithm 1 using ClippedGD finds a meta-parameter $\boldsymbol{\theta}$ satisfying $\mathbb{E}[\|\nabla F(\boldsymbol{\theta})\|] \leq \varepsilon + b$ in:*

- ***Deterministic Case** ($\tilde{\sigma} = 0$):* $\mathscr{O}\left(\frac{\mathscr{L}_0\Delta}{\varepsilon^2} + \frac{\mathscr{L}_1^2\Delta}{\mathscr{L}_0}\right)$ *outer steps.*

- ***Stochastic Case** ($\tilde{\sigma} > 0$):* $\mathscr{O}\left(\Delta\tilde{\sigma}^2 \max\left(\frac{\mathscr{L}_0}{\varepsilon^4}, \frac{\mathscr{L}_1^4}{\mathscr{L}_0^3}\right)\right)$ *outer steps.*

*where $b$ is the estimator bias: $b = \mathscr{O}(\lambda\sqrt{\delta})$ for the forward approximation and $b = \mathscr{O}(\lambda\delta^{2/3})$ for the symmetric approximation.*

---

**Theorem 2 (FO-B-MAML using SGD)** *Under the classical smoothness (Proposition 3), Algorithm 1 using SGD finds a meta-parameter $\boldsymbol{\theta}$ satisfying $\mathbb{E}[\|\nabla F(\boldsymbol{\theta})\|] \leq \varepsilon + b$ in $\mathscr{O}\left(\frac{\mathscr{L}\Delta}{\varepsilon^2} + \frac{\mathscr{L}\Delta\tilde{\sigma}^2}{\varepsilon^4}\right)$ outer steps.*

---

**Complexity Comparison and Total Cost.** The total complexity (total gradient calls) is $M \times Outer\text{-}steps(\varepsilon) \times Inner\text{-}steps(\delta)$. In the stochastic regime, the inner solver (e.g., SGD) requires $Inner\text{-}steps = \mathscr{O}(\sigma^2\delta^{-1})$.

**Stochastic Case Leading Terms.** To reach $\varepsilon$-stationarity, we require the bias $b \sim \varepsilon$. This leads to the following leading-order complexities in the stochastic regime:

- **Forward Estimator**: Requiring $\sqrt{\delta} \sim \varepsilon \implies \delta \sim \varepsilon^2$. The inner complexity is $\mathscr{O}(\sigma^2\varepsilon^{-2})$. Multiplying by the outer complexity $\mathscr{O}(\tilde{\sigma}^2\varepsilon^{-4})$, the leading term is $\mathscr{O}(M\Delta\tilde{\sigma}^2\sigma^2\varepsilon^{-6})$.

- **Symmetric Estimator**: Requiring $\delta^{2/3} \sim \varepsilon \implies \delta \sim \varepsilon^{1.5}$. The inner complexity is reduced to $\mathscr{O}(\sigma^2\varepsilon^{-1.5})$. The total leading term becomes $\mathscr{O}(M\Delta\tilde{\sigma}^2\sigma^2\varepsilon^{-5.5})$.

The symmetric approximation provides a significant theoretical advantage, saving a factor of $\varepsilon^{-0.5}$ in total complexity compared to the forward approach. These accelerated symmetric results are new and provide a strict improvement over previous MAML theory.

Table 1: Comparison of gradient-based meta-learning methods (deterministic setting, $M = 1$). Total complexity is the product of inner and outer iterations to reach $\varepsilon$-stationarity.

| Algorithm | Compute (Total) | Memory | First-order | Bias |
|---|---|---|---|---|
| MAML | $\mathscr{O}(\varepsilon^{-2}\sqrt{\kappa}\log(1/\varepsilon))$ | $\text{Mem(activations)} \cdot \sqrt{\kappa}$ | No | $\delta$ |
| iMAML | $\mathscr{O}(\varepsilon^{-2}\sqrt{\kappa}\log(1/\varepsilon))$ | $\text{Mem(activations)}$ | No | $\delta$ |
| FO-MAML/Reptile | $\mathscr{O}(\varepsilon^{-2}\sqrt{\kappa}\log(1/\varepsilon))$ | $\text{Mem}(\boldsymbol{\theta})$ | Yes | 1 |
| FO-B-MAML (For) | $\mathscr{O}(\varepsilon^{-2}\sqrt{\kappa}\log(1/\varepsilon))$ | $\text{Mem}(\boldsymbol{\theta})$ | Yes | $\sqrt{\delta}$ |
| FO-B-MAML (Sym) | $\mathscr{O}(\varepsilon^{-2}\sqrt{\kappa}\log(1/\varepsilon))$ | $\text{Mem}(\boldsymbol{\theta})$ | Yes | $\delta^{2/3}$ |

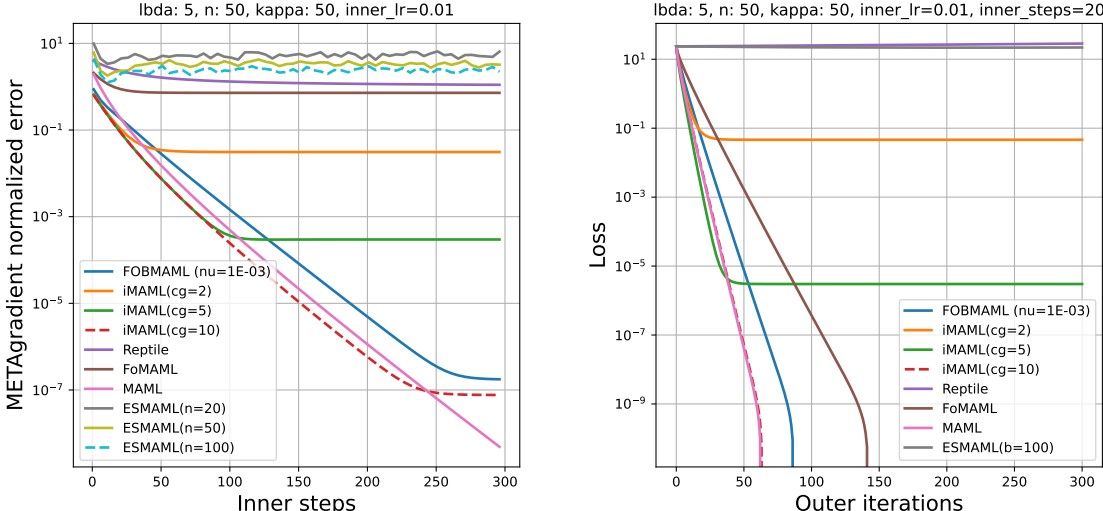

Figure 1: **(Above)** Quality of the meta-gradient estimates as a function of inner steps. **(Below)** Outer loss as a function of iterations for 20 inner steps.

**Discussion on Memory Efficiency: Activations vs. Parameters.** A critical distinction between optimization-based meta-learning algorithms lies in their memory overhead. Traditional methods like MAML and iMAML rely on second-order information, which necessitates storing the **activations** of the inner-loop trajectory or the computation graph for implicit differentiation. For deep networks with high-dimensional activation maps, activation storage is the primary bottleneck, often far exceeding the memory required for model parameters.

In contrast, **FO-B-MAML** only requires the calculation of point estimates $\tilde{\phi}_{i,v}(\boldsymbol{\theta})$, offering two primary benefits:

- **Parameter-Only Storage**: FO-B-MAML relies on $\mathrm{Mem}(\boldsymbol{\theta})$. While the symmetric approximation involves two subproblems ($\tilde{\phi}_{i,v}$ and $\tilde{\phi}_{i,-v}$), we avoid a $2\times$ parameter overhead by solving these subproblems **sequentially**. By computing the first estimate and then accumulating the second, the peak memory footprint remains identical to standard first-order training.

- **Scalability**: This allows FO-B-MAML to scale to activation-heavy architectures where MAML would typically trigger Out-of-Memory (OOM) errors, as evidenced by our ConvNet experiments.

## 6 Experiments

We evaluate Algorithm 1 across three distinct settings: a synthetic linear regression problem to measure meta-gradient quality, an MNIST-1D benchmark for classification performance, and a CNN scalability test to verify memory efficiency on deep architectures. We compare our results against iMAML, MAML, Reptile, and FO-MAML.

### 6.1 Meta-Gradient Quality and Convergence

We first consider a synthetic linear regression problem which allows for a closed-form meta-gradient calculation. Figure 1 illustrates that FO-B-MAML's meta-gradient approximation benefits continuously from increased inner steps (decreasing $\delta$), a property not shared by FO-MAML or Reptile. FO-B-MAML is highly competitive with iMAML; notably, it outperforms iMAML when the latter is restricted to a small number of conjugate gradient steps ($cg \in \{2, 5\}$).

## 6.2 MNIST-1D Classification

To assess performance on non-linear tasks, we utilize the MNIST-1D dataset in a 3-way 5-shot meta-learning setup. As shown in Figure 2, FO-B-MAML tracks the performance of second-order MAML much more closely than other first-order methods. Specifically, FO-B-MAML reaches an accuracy above 0.85 within 100 iterations and maintains a final accuracy competitive with MAML (approx. 0.95) while using only first-order information. In contrast, FO-MAML remains below 0.55 accuracy throughout the training.

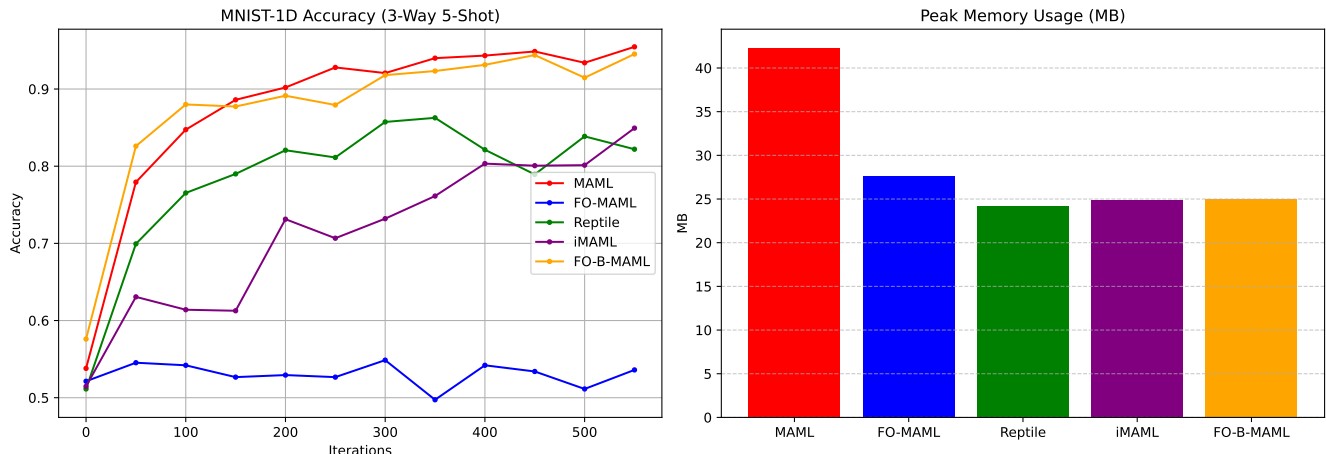

Figure 2: MNIST-1D 3-Way 5-Shot accuracy comparison across 600 iterations.

## 6.3 Omniglot Few-Shot Classification

To address the requirement for evaluation on standard, higher-dimensional benchmarks, we evaluate FO-B-MAML on the Omniglot dataset (Lake et al., 2015). This dataset contains handwritten characters from 50 different alphabets, presenting a complex 2D image challenge. We test our method in the standard 5-way 1-shot and 5-way 5-shot settings.

**Computational and Sample Efficiency.** FO-B-MAML achieves competitive performance while maintaining high computational efficiency. A key advantage of our method is its superior inner-loop economy: we utilize only 5 adaptation steps per task, whereas the iMAML (GD) baseline requires 16 steps to solve 5-way tasks (Rajeswaran et al., 2019b). While our reported results are achieved within a relatively short meta-training schedule, we anticipate that further performance gains could be obtained by training for a larger number of outer iterations.

Table 2: Omniglot Results. FO-B-MAML results are averaged over 5 seeds. Baseline results for MAML, FO-MAML, Reptile, and iMAML are taken from (Finn et al., 2017a; Nichol et al., 2018; Rajeswaran et al., 2019b).

| Algorithm | 5-way 1-shot | 5-way 5-shot |
|---|---|---|
| MAML (Finn et al., 2017a) | $98.7 \pm 0.4\%$ | $\mathbf{99.9 \pm 0.1}\%$ |
| FO-MAML (Finn et al., 2017a) | $98.3 \pm 0.5\%$ | $99.2 \pm 0.2\%$ |
| Reptile (Nichol et al., 2018) | $97.68 \pm 0.04\%$ | $99.48 \pm 0.06\%$ |
| iMAML, GD (Rajeswaran et al., 2019b) | $99.16 \pm 0.35\%$ | $99.67 \pm 0.12\%$ |
| iMAML, Hessian-Free (Rajeswaran et al., 2019b) | $\mathbf{99.50 \pm 0.26}\%$ | $99.74 \pm 0.11\%$ |
| **FO-B-MAML (Ours)** | $\mathbf{99.24 \pm 0.29}\%$ | $\mathbf{99.70 \pm 0.17}\%$ |

**Analysis and Fairness of Comparison.** As shown in Table 2, FO-B-MAML matches or exceeds the accuracy of iMAML (GD) despite using $3\times$ fewer inner iterations. We emphasize that iMAML (GD) represents the most direct baseline for our method, as both algorithms employ Gradient Descent (GD) as the inner-task solver. This parity suggests

that our perturbed bi-level gradient estimation provides a high-quality update signal that facilitates rapid adaptation even with a coarse inner-loop solution.

While iMAML (Hessian-Free) reports a marginally higher 1-shot score, that variant utilizes a second-order Newton CG method for inner-loop optimization. A fair comparison with Hessian-free strategies would necessitate upgrading FO-B-MAML's inner solver to a similar second-order algorithm. Our results demonstrate that FO-B-MAML enables a simple first-order GD inner solver to achieve performance previously associated with implicit methods using more complex, higher-order optimizers.

Furthermore, it is important to note that the performance reported for FO-B-MAML was obtained by only conducting a hyperparameter search on the meta-learning rate (outer step size). Given the sensitivity of meta-learning algorithms to hyperparameters such as the perturbation scale $\nu$ (set to 0.1) and the regularization parameter $\lambda$ (set to 2.0), it is highly probable that a more exhaustive search across the entire hyperparameter space would yield even higher accuracies. The fact that FO-B-MAML achieves competitive results with minimal tuning highlights its robustness and practical utility for large-scale applications where extensive searching is computationally prohibitive.

### 6.4 Memory Scalability: Transformers and Deep CNNs

A core contribution of FO-B-MAML is its ability to scale to deep, activation-heavy architectures where traditional methods fail due to the "activation bottleneck." We measured peak memory usage on two distinct architectures: an activation-heavy ConvNet by varying the number of channels, and a Transformer block by varying the embedding dimension ($d_{model}$).

**The Activation Bottleneck.** Traditional methods like MAML and iMAML rely on second-order information, which necessitates storing the **activations** of the inner-loop trajectory or the full computation graph for backpropagation. In Transformers, the quadratic complexity of the self-attention mechanism generates massive activation maps proportional to the square of the sequence length ($Seq \times Seq$). As shown in our benchmarks, MAML's memory usage grows aggressively with model width. For the MNIST-1D task, MAML requires over 40 MB, while FO-B-MAML requires only 25 MB.

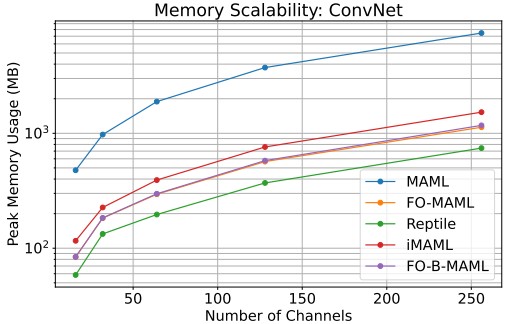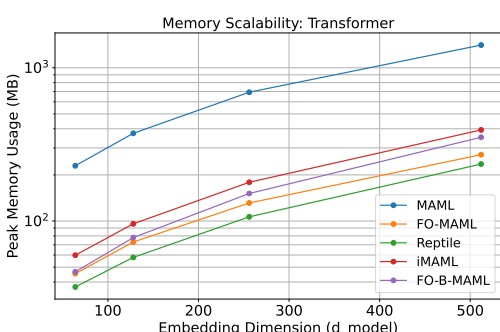

Figure 3: Peak memory usage comparison (MB) as a function of ConvNet channels (left) and Transformer embedding dimension (right). FO-B-MAML maintains a significantly flatter memory footprint compared to second-order methods.

In contrast, **FO-B-MAML** only requires the calculation of parameter point-estimates $\tilde{\phi}_{i,\nu}(\boldsymbol{\theta})$. By solving the perturbed subproblems sequentially, it discards intermediate activations—such as massive attention maps—before proceeding to the next estimate. As seen in Figure 3, FO-B-MAML maintains a memory footprint around $10^3$ MB as ConvNet channel counts increase to 250, and stays below $3 \times 10^2$ MB for Transformer dimensions up to 512. Meanwhile, MAML's requirements scale steeply toward significantly higher values, often triggering Out-of-Memory (OOM) errors. This confirms the practical utility of our method for modern, activation-heavy architectures.

## 7 General Discussion

**Extension.** In this work, we focused on the specific regularization form in (6) and (5). However, our approach is inherently general and can be extended to meta-learn other hyperparameters or shared components, such as common

decoders. For example, consider a general inner problem:

$$\phi_i^\star \left( \boldsymbol{h} := \{ \boldsymbol{\theta}, \boldsymbol{\theta}_d, \lambda \} \right) = \arg\min_{\boldsymbol{\phi}} \left[ \hat{f}_i(\boldsymbol{\theta}_d, \boldsymbol{\phi}) + \lambda \mathscr{R}eg(\boldsymbol{\phi}; \boldsymbol{\theta}) \right] , \tag{20}$$

where $\boldsymbol{\theta}_d$ denotes shared parameters. By adding a perturbation $\nu f(\boldsymbol{\phi})$, the meta-gradient satisfies $\nabla F_i(\boldsymbol{h}) = \frac{d \partial_{\boldsymbol{h}} g(\phi_{i,\nu}^\star(\boldsymbol{h}), \boldsymbol{h})}{d\nu}\Big|_{\nu=0}$. This allows for high-precision meta-gradients in complex architectures without second-order derivatives.

**Practical Impact of Memory Efficiency.** Our experimental results on deep CNNs and Transformers highlight that FO-B-MAML bypasses the "activation bottleneck" inherent in second-order methods. While MAML requires memory that scales steeply with model width and sequence length due to the accumulation of high-dimensional activation maps within the computation graph—particularly the quadratic $\mathscr{O}(L^2)$ cost of self-attention—FO-B-MAML relies only on parameter point-estimates.

By solving perturbed subproblems sequentially and discarding intermediate activations before proceeding to the next estimate, we maintain a memory footprint nearly identical to standard first-order training. This enables meta-learning on modern, activation-heavy architectures and long-sequence tasks that would otherwise trigger Out-of-Memory (OOM) errors in second-order frameworks.

**Limitations.** A primary drawback of FO-B-MAML is the increased computational cost in the inner loop, which requires solving the task optimization problem at least twice to compute the finite-difference estimate. While this overhead is partially offset by the absence of second-order derivative calculations and the potential for parallelizing subproblem solutions, reducing this cost remains an active area of research. One potential alternative to finite differences is to apply automatic differentiation (AD) directly to the scalar perturbation parameter $\nu$ to recover the meta-gradient in a single pass. However, such an approach would require backpropagating through the entire inner-loop trajectory, which re-introduces the "activation bottleneck" and incurs memory costs comparable to vanilla MAML.

Furthermore, our framework's theoretical robustness relies on the assumption of local strong convexity. While the proximal regularization $\lambda$ helps enforce this condition and stabilize the inner loop, it introduces a tuning dependency. It is worth noting that both $\lambda$ and $\nu$ could theoretically be updated online using their respective meta-gradients—calculated via the same perturbation identity—to automate their selection. However, the empirical validation of such an auto-tuning schedule is left for future work. For now, the algorithm's performance remains sensitive to these values, necessitating the use of the practical "Step-Aware" rules and diagnostic strategies detailed in Appendix A.

# 8 Conclusion

We introduced FO-B-MAML, a first-order meta-learning algorithm derived from a bi-level optimization perspective. Our framework includes a symmetric estimator that achieves an improved $\mathscr{O}(\delta^{2/3})$ bias rate. Theoretical analysis confirms that when paired with Stochastic Normalized Gradient Descent with Momentum (SNGDM), FO-B-MAML achieves superior convergence compared to traditional first-order methods.

Experimentally, we demonstrated that our method achieves high accuracy on MNIST-1D, tracking closely with second-order MAML, while maintaining the memory efficiency required for deep, activation-heavy CNNs. FO-B-MAML thus provides a robust, high-precision alternative for large-scale meta-learning where second-order information is computationally or memory-prohibitive.

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

# A  Empirical Analysis of Hyperparameter Sensitivity

To provide practical guidance on hyperparameter selection and address concerns regarding auto-tuning, we conducted a series of controlled experiments using the synthetic quadratic objective defined in Section 3.3. These experiments isolate the mathematical behavior of the **Symmetric FO-B-MAML** estimator by comparing it against analytical meta-gradients.

## A.1  The "Step-Aware" Rule for Perturbation Scale ($\nu$)

The selection of $\nu$ involves a fundamental trade-off between numerical stability and approximation bias. As demonstrated in Figure 4, the optimal $\nu$ is intrinsically linked to the inner solver's precision $\delta$, which is a function of the number of inner steps $K$.

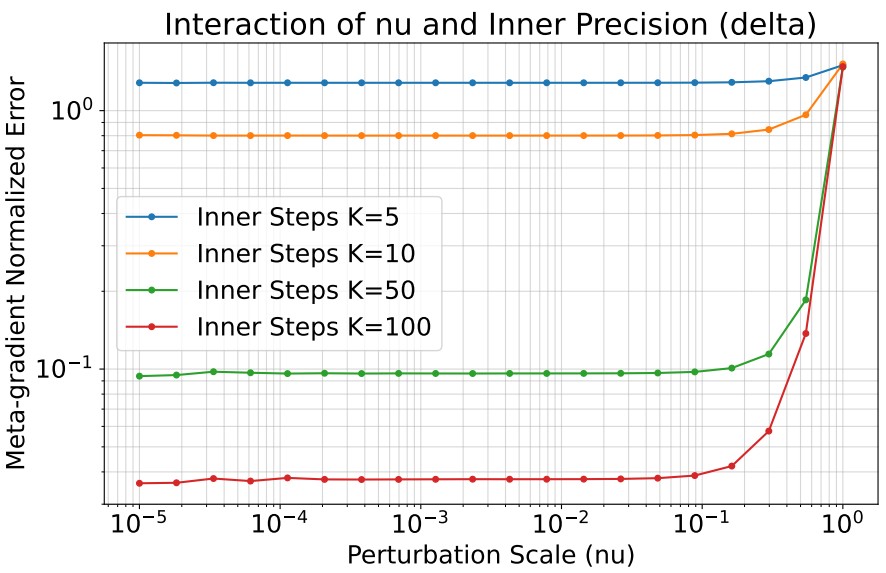

Figure 4: **FO-B-MAML Estimator Error**: Meta-gradient normalized error as a function of perturbation scale $\nu$ for different inner precision levels $K$. Increased inner steps $K$ significantly reduce the baseline error across all tested scales.

- **Numerical Instability**: For small values of $\nu$ (e.g., $10^{-5}$ to $10^{-2}$), the meta-gradient error remains relatively flat or high because it is dominated by the solver precision $\delta$. Dividing by an excessively small $\nu$ amplifies the numerical "noise" of the inexact inner solution.

- **Optimal Elbow Points**: As the number of inner steps $K$ increases (reducing $\delta$), the meta-gradient normalized error drops significantly across all tested scales. For $K = 100$, the estimator achieves an error rate near $10^{-2}$, whereas $K = 5$ remains an order of magnitude higher.

- **Bias Threshold**: Once $\nu$ exceeds $10^{-1}$, the error spikes sharply for all $K$, reflecting the $O(\nu^2)$ bias of the symmetric estimator becoming the primary source of error.

**Practical Rule of Thumb:**

- **Coarse Adaptation** ($K \leq 10$ **steps**): Use a larger scale $\nu \in [0.1, 0.5]$ to ensure the meta-gradient signal is not lost to solver noise.

- **High-Precision Adaptation** ($K \geq 50$ **steps**): Practitioners can safely decrease $\nu$ to 0.01 or 0.001 to minimize approximation bias and fully leverage the improved $O(\delta^{2/3})$ bias rate.

### A.2 Analysis of Regularization Strength ($\lambda$)

The parameter $\lambda$ controls the trade-off between the meta-prior and task-specific data. In our quadratic setting, the task-specific solution is $\phi_i^*(\theta) = (A_i + \lambda I)^{-1}(\lambda \theta - b_i)$, showing that $\lambda$ directly scales the influence of the initialization $\theta$.

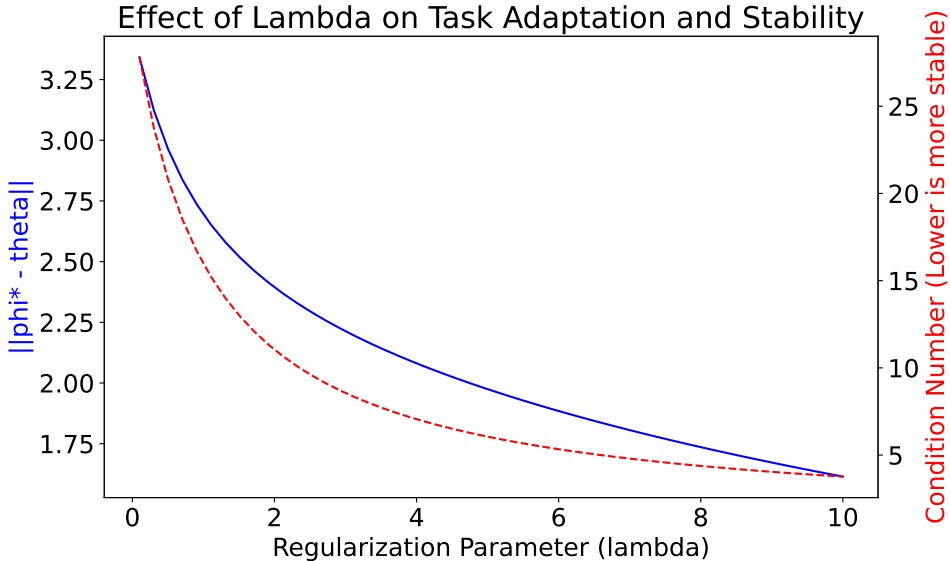

Figure 5: **Effect of Lambda on Task Adaptation and Stability**: $\lambda$ acts as a trade-off between the distance of adaptation (blue line) and the numerical stability of the inner optimization (red dashed line).

- **Adaptation Trade-off**: At low $\lambda$ (e.g., 0.1), the adaptation distance $\|\phi^* - \theta\|$ is maximized (approx. 3.25), indicating the model relies heavily on task data. As $\lambda$ increases to 10, the distance drops significantly (below 1.75), creating a "stiff" prior that favors the meta-initialization.

- **Numerical Stability**: Increasing $\lambda$ dramatically improves (lowers) the condition number of the inner-optimization matrix from approximately 25 down to 5. This stabilizes the inner loop and ensures a unique, well-behaved solution path for the perturbed problem.

**Practical Guidance:**

- **Manual Selection**: Start with $\lambda \approx 1/\alpha$, where $\alpha$ is the standard inner learning rate. We found $\lambda = 2.0$ to be robust across benchmarks.

- **Meta-Learning** $\lambda$: As discussed in Section 7, $\lambda$ can be treated as a meta-parameter. Our framework allows estimating $\nabla_\lambda F$ using the same symmetric-difference identity, enabling the model to learn optimal "adaptation stiffness" automatically.

## B  Additional experiment: Memory Scalability in Attention-Based Architectures

To investigate the performance of FO-B-MAML on modern, high-dimensional architectures, we evaluate its memory footprint against standard meta-learning baselines using a Transformer backbone. Unlike convolutional or fully connected networks, Transformers utilize a self-attention mechanism that incurs a quadratic memory cost $O(L^2)$ relative to the input sequence length $L$. In a meta-learning context, this cost is compounded for second-order methods that must store these large activation maps across multiple inner-loop adaptation steps.

**The Activation Bottleneck.** Figure 6 illustrates the peak memory usage as a function of sequence length $L$ while maintaining a fixed embedding dimension of $Dim = 128$.

- **MAML**: Exhibits an aggressive growth curve, surpassing $10^3$ MB as $L$ reaches 512. This is due to the accumulation of high-dimensional attention activations within the computation graph required for the second-order meta-gradient.

- **iMAML**: Shows improved efficiency over vanilla MAML but still maintains a consistently higher memory profile than first-order or perturbed alternatives.

- **FO-B-MAML**: Tracks closely with first-order methods like Reptile and FO-MAML. By utilizing perturbed point-estimates rather than backpropagating through the optimization path, FO-B-MAML successfully avoids the quadratic growth associated with maintaining an inner-loop computation graph.

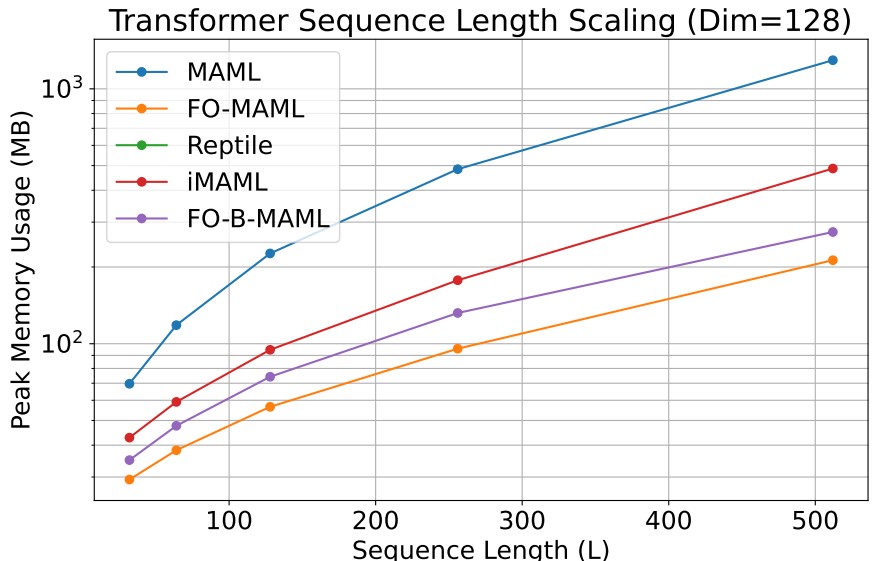

Figure 6: Peak memory usage (MB) for Transformer sequence length scaling ($Dim = 128$). While MAML approaches $10^3$ MB at $L = 512$, FO-B-MAML maintains a significantly lower memory footprint, comparable to first-order methods.

**Conclusion on Scalability.** The results confirm that FO-B-MAML is uniquely suited for scaling meta-learning to long-sequence tasks. By decoupling the meta-gradient estimation from the activation-heavy optimization trajectory, our method enables the use of sophisticated attention-based models on hardware where second-order MAML would otherwise trigger Out-of-Memory (OOM) errors.

## C  Missing proofs

### C.1  Proof of Proposition 1

The fact that $v \mapsto \phi_{i,v}^\star(\boldsymbol{\theta})$ is differentiable at $v = 0$ will be proven later.

We have that

$$\phi_{i,v}^\star(\boldsymbol{\theta}) \in Arg\min_{\boldsymbol{\phi}} v f_i(\boldsymbol{\phi}) + \hat{f}_i(\boldsymbol{\phi}) + \frac{\lambda}{2}\|\boldsymbol{\phi} - \boldsymbol{\theta}\|^2$$

This means

$$v\nabla f_i(\phi_{i,v}^\star(\boldsymbol{\theta})) + \nabla\hat{f}_i(\phi_{i,v}^\star(\boldsymbol{\theta})) + \lambda(\phi_{i,v}^\star(\boldsymbol{\theta}) - \boldsymbol{\theta}) = \mathbf{0} \tag{21}$$

Taking the derivative of the above equation with respect to $\nu$ gives

$$\nabla f_i(\phi_{i,\nu}^\star(\boldsymbol{\theta})) + \nu \nabla^2 f_i(\phi_{i,\nu}^\star(\boldsymbol{\theta}))\frac{d\phi_{i,\nu}^\star(\boldsymbol{\theta})}{d\nu} + \nabla^2 \hat{f}_i(\phi_{i,\nu}^\star(\boldsymbol{\theta}))\frac{d\phi_{i,\nu}^\star(\boldsymbol{\theta})}{d\nu} + \lambda\frac{d\phi_{i,\nu}^\star(\boldsymbol{\theta})}{d\nu} = 0$$

Which yields the following expression

$$\frac{d\phi_{i,\nu}^\star(\boldsymbol{\theta})}{d\nu} = -\Big(\nu\nabla^2 f_i(\phi_{i,\nu}^\star(\boldsymbol{\theta})) + \nabla^2 \hat{f}_i(\phi_{i,\nu}^\star(\boldsymbol{\theta})) + \lambda\boldsymbol{I}\Big)^{-1}\nabla f_i(\phi_{i,\nu}^\star(\boldsymbol{\theta}))$$

We set $\nu = 0$ and get:

$$\frac{d\phi_{i,\nu}^\star(\boldsymbol{\theta})}{d\nu}\Big|_{\nu=0} = -\Big(\nabla^2 \hat{f}_i(\phi_i^\star(\boldsymbol{\theta})) + \lambda\boldsymbol{I}\Big)^{-1}\nabla f_i(\phi_i^\star(\boldsymbol{\theta})) = -\frac{1}{\lambda}\nabla F_i(\boldsymbol{\theta})$$

## C.2 Proof of the Bias of the Forward Approximation

Using Equation (21), we have:

$$\phi_{i,\nu}^\star(\boldsymbol{\theta}) = \boldsymbol{\theta} - \frac{1}{\lambda}\Big(\nu\nabla f_i(\phi_{i,\nu}^\star(\boldsymbol{\theta})) + \nabla\hat{f}_i(\phi_{i,\nu}^\star(\boldsymbol{\theta}))\Big) \tag{22}$$

Thus

$$\lambda\frac{\phi_{i,0}^\star(\boldsymbol{\theta}) - \phi_{i,\nu}^\star(\boldsymbol{\theta})}{\nu} = \nabla f_i(\phi_{i,\nu}^\star(\boldsymbol{\theta})) + \frac{\nabla\hat{f}_i(\phi_{i,\nu}^\star(\boldsymbol{\theta})) - \nabla\hat{f}_i(\phi_{i,0}^\star(\boldsymbol{\theta}))}{\nu} . \tag{23}$$

The form of the meta-gradient in Equation 7 implies that:

$$\nabla F_i(\boldsymbol{\theta}) = \nabla f_i(\phi_{i,0}^\star(\boldsymbol{\theta})) - \frac{1}{\lambda}\nabla^2 \hat{f}_i(\phi_{i,0}^\star(\boldsymbol{\theta}))\nabla F_i(\boldsymbol{\theta}). \tag{24}$$

(24) – (23) gives :

$$\nabla F_i(\boldsymbol{\theta}) - \boldsymbol{g}_{i,\nu}^{\text{For}} = \underbrace{\nabla f_i(\phi_{i,0}^\star(\boldsymbol{\theta})) - \nabla f_i(\phi_{i,\nu}^\star(\boldsymbol{\theta}))}_{\textbf{(I)}}$$

$$+ \underbrace{\frac{\nabla\hat{f}_i(\phi_{i,\nu}^\star(\boldsymbol{\theta})) - \nabla\hat{f}_i(\phi_{i,0}^\star(\boldsymbol{\theta}))}{\nu} - \frac{1}{\lambda}\nabla^2 \hat{f}_i(\phi_{i,0}^\star(\boldsymbol{\theta}))\nabla F_i(\boldsymbol{\theta})}_{\textbf{(II)}}.$$

Let's define $\mathscr{B}_\nu = \|\nabla F_i(\boldsymbol{\theta}) - \boldsymbol{g}_{i,\nu}^{\text{For}}\|$ the bias of the forward approximation. The norm of the first term **(I)** can be easily bounded using the smoothness of $f_i$ by:

$$\|\textbf{(I)}\| \le L_1\|\phi_{i,\nu}^\star(\boldsymbol{\theta}) - \phi_{i,0}^\star(\boldsymbol{\theta})\|.$$

The second term **(II)** can be simplified using Cauchy's theorem which guarantees the existence of $\phi$ such that

$$\frac{\nabla\hat{f}_i(\phi_{i,\nu}^\star(\boldsymbol{\theta})) - \nabla\hat{f}_i(\phi_{i,0}^\star(\boldsymbol{\theta}))}{\nu} = \nabla^2 \hat{f}_i(\phi_{i,0}^\star(\boldsymbol{\theta}))\Big[\frac{\phi_{i,\nu}^\star(\boldsymbol{\theta}) - \phi_{i,0}^\star(\boldsymbol{\theta})}{\nu}\Big] + \frac{1}{2\nu}\nabla^3 \hat{f}_i(\phi_{i,0}^\star(\boldsymbol{\theta}))\Big[\phi_{i,\nu}^\star(\boldsymbol{\theta}) - \phi_{i,0}^\star(\boldsymbol{\theta})\Big]^2,$$

Thus, using Assumptions 1 and 3, we get:

$$\|\textbf{(II)}\| \le \frac{\hat{L}_1}{\lambda}\mathscr{B}_\nu + \frac{\hat{L}_2}{2\nu}\|\phi_{i,\nu}^\star(\boldsymbol{\theta}) - \phi_{i,0}^\star(\boldsymbol{\theta})\|^2.$$

Overall,

$$\mathscr{B}_\nu \leq L_1 \|\phi_{i,\nu}^\star(\boldsymbol{\theta}) - \phi_{i,0}^\star(\boldsymbol{\theta})\| + \frac{\hat{L}_1}{\lambda}\mathscr{B}_\nu + \frac{\hat{L}_2}{2\nu}\|\phi_{i,\nu}^\star(\boldsymbol{\theta}) - \phi_{i,0}^\star(\boldsymbol{\theta})\|^2 \, ,$$

Which implies :

$$(1 - \frac{\hat{L}_1}{\lambda})\mathscr{B}_\nu \leq \|\phi_{i,\nu}^\star(\boldsymbol{\theta}) - \phi_{i,0}^\star(\boldsymbol{\theta})\|\left(L_1 + \frac{\hat{L}_2}{2\nu}\|\phi_{i,\nu}^\star(\boldsymbol{\theta}) - \phi_{i,0}^\star(\boldsymbol{\theta})\|\right). \tag{25}$$

All that is left is to bound the term $\|\phi_{i,\nu}^\star(\boldsymbol{\theta}) - \phi_{i,0}^\star(\boldsymbol{\theta})\|$.

If we go back to (22), then we can write:

$$\begin{aligned}\|\phi_{i,\nu}^\star(\boldsymbol{\theta}) - \phi_{i,0}^\star(\boldsymbol{\theta})\| &= \|\frac{1}{\lambda}\left(\nabla\hat{f}_i(\phi_{i,0}^\star(\boldsymbol{\theta})) - \nabla\hat{f}_i(\phi_{i,\nu}^\star(\boldsymbol{\theta})) + \nu\nabla f_i(\phi_{i,\nu}^\star(\boldsymbol{\theta}))\right)\| \\ &\leq \frac{\hat{L}_1}{\lambda}\|\phi_{i,\nu}^\star(\boldsymbol{\theta}) - \phi_{i,0}^\star(\boldsymbol{\theta})\| + \frac{\nu L_0}{\lambda}.\end{aligned}$$

Thus:

$$(1 - \frac{\hat{L}_1}{\lambda})\|\phi_{i,\nu}^\star(\boldsymbol{\theta}) - \phi_{i,0}^\star(\boldsymbol{\theta})\| \leq \frac{\nu L_0}{\lambda}.$$

For simplicity we assume, $\lambda \geq 2\hat{L}_1$ which gives:

$$\|\phi_{i,\nu}^\star(\boldsymbol{\theta}) - \phi_{i,0}^\star(\boldsymbol{\theta})\| \leq \frac{2\nu L_0}{\lambda} \, . \tag{26}$$

Plugging the result in Eq (26) and using $\lambda \geq 2\hat{L}_1$ gives:

$$\mathscr{B}_\nu \leq \frac{L_0}{\lambda}\left(L_1 + \frac{L_0\hat{L}_2}{\lambda}\right)\nu.$$

The overall bias resulting from using approximations of $\phi_{i,\nu}^\star(\boldsymbol{\theta})$ instead of their exact values is bounded by:

$$\frac{L_0}{\lambda}\left(L_1 + \frac{L_0\hat{L}_2}{\lambda}\right)\nu + \frac{2\delta}{\nu}, \tag{27}$$

which is minimized for $\nu = \sqrt{\dfrac{2\lambda^2\delta}{L_0(L_1\lambda + L_0\hat{L}_2)}}$, using this value of $\nu$ in the overall bias (27), gives the result of Corrolary 1.

## C.3 Proof of Improved Bias Rate for the Symmetric Estimator

To achieve the results for the symmetric estimator $g_{i,\nu}^{\text{Sym}}$, we adopt strengthened regularity conditions:

- **Assumption 1':** $\hat{f}_i$ is $C^4$ with derivatives $\nabla^3 \hat{f}_i$ and $\nabla^4 \hat{f}_i$ bounded by $\hat{L}_2$ and $\hat{L}_3$ respectively.

- **Assumption 2':** $f_i$ is $C^3$ with derivatives $\nabla^2 f_i$ and $\nabla^3 f_i$ bounded by $L_1$ and $L_2$ respectively.

- **Assumption 3:** The inner objective is $\mu$-strongly convex.

### C.3.1 Bounding the Solution Path Derivatives

Let $z(\nu) = \phi_{i,\nu}^\star(\boldsymbol{\theta})$ be the solution to the perturbed inner problem. Differentiating the stationary condition (21) with respect to $\nu$:

$$A(z,\nu)z'(\nu) + \nabla f_i(z) = \mathbf{0}, \tag{28}$$

where $A(z,v) = v\nabla^2 f_i(z) + \nabla^2 \hat{f}_i(z) + \lambda I$. By Assumption 3, $\|A^{-1}\| \leq \frac{1}{\mu}$. Thus, $\|z'(v)\| \leq \frac{L_0}{\mu}$. Differentiating (28) again with respect to $v$:

$$A z'' + \left[\nabla^2 f_i(z) + (v\nabla^3 f_i(z) + \nabla^3 \hat{f}_i(z))z'\right]z' + \nabla^2 f_i(z)z' = 0. \tag{29}$$

Rearranging for $z''$ and taking the norm:

$$\|z''\| \leq \frac{1}{\mu}\left[2L_1\frac{L_0}{\mu} + (vL_2 + \hat{L}_2)\left(\frac{L_0}{\mu}\right)^2\right] = \mathscr{C}_2. \tag{30}$$

Under Assumptions 1' and 2', differentiating a third time ensures the existence of a constant $\mathscr{C}_3$ such that $\sup_{\xi \in [-v,v]}\|z'''(\xi)\| \leq \mathscr{C}_3$.

### C.3.2 Symmetric Estimator Bias Bound

Using the symmetric difference quotient and Taylor's theorem:

$$z(v) = z(0) + vz'(0) + \frac{v^2}{2}z''(0) + \frac{v^3}{6}z'''(\xi_1), \quad \xi_1 \in [0,v]$$

$$z(-v) = z(0) - vz'(0) + \frac{v^2}{2}z''(0) - \frac{v^3}{6}z'''(\xi_2), \quad \xi_2 \in [-v,0].$$

The symmetric estimator is defined as $g_{i,v}^{\text{Sym}} = -\lambda\frac{z(v)-z(-v)}{2v}$. Substituting the expansions:

$$g_{i,v}^{\text{Sym}} = -\lambda z'(0) - \frac{\lambda v^2}{12}\left(z'''(\xi_1) + z'''(\xi_2)\right). \tag{31}$$

Since $\nabla F_i(\theta) = -\lambda z'(0)$ (21), the approximation bias is bounded by $\frac{\lambda v^2 \mathscr{C}_3}{6}$. Including the solver precision $\delta$, the total error $E(v)$ is:

$$E(v) \leq \frac{\lambda v^2 \mathscr{C}_3}{6} + \frac{\lambda\delta}{v}. \tag{32}$$

Minimizing for $v$ yields $v^\star = \sqrt[3]{3\delta/\mathscr{C}_3}$, which leads to the final bias bound:

$$\text{Bias}_{\text{Sym}} \leq \frac{\lambda}{2}(3\delta)^{2/3}\mathscr{C}_3^{1/3} = \mathcal{O}(\lambda\delta^{2/3}). \tag{33}$$

This confirms that the symmetric approximation provides a superior bias rate compared to the forward approximation.

### C.4 Proof of Proposition 3

Let $\theta, \theta' \in \Theta$. Using Eq (24), we have:

$$\nabla F_i(\theta) = \nabla f_i(\phi_{i,0}^\star(\theta)) - \frac{1}{\lambda}\nabla^2 \hat{f}_i(\phi_{i,0}^\star(\theta))\nabla F_i(\theta)$$

$$\nabla F_i(\theta') = \nabla f_i(\phi_{i,0}^\star(\theta')) - \frac{1}{\lambda}\nabla^2 \hat{f}_i(\phi_{i,0}^\star(\theta'))\nabla F_i(\theta')$$

$$\begin{aligned}
\nabla F_i(\theta) - \nabla F_i(\theta') &= \nabla f_i(\phi_{i,0}^\star(\theta)) - \nabla f_i(\phi_{i,0}^\star(\theta')) \\
&\quad + \frac{1}{\lambda}\left(\nabla^2 \hat{f}_i(\phi_{i,0}^\star(\theta'))\nabla F_i(\theta') - \nabla^2 \hat{f}_i(\phi_{i,0}^\star(\theta))\nabla F_i(\theta)\right) \\
&= \nabla f_i(\phi_{i,0}^\star(\theta)) - \nabla f_i(\phi_{i,0}^\star(\theta')) \\
&\quad + \frac{1}{\lambda}\left(\nabla^2 \hat{f}_i(\phi_{i,0}^\star(\theta')) - \nabla^2 \hat{f}_i(\phi_{i,0}^\star(\theta))\right)\nabla F_i(\theta') \\
&\quad - \frac{1}{\lambda}\nabla^2 \hat{f}_i(\phi_{i,0}^\star(\theta))\left(\nabla F_i(\theta) - \nabla F_i(\theta')\right)
\end{aligned}$$

Thus:

$$\|\nabla F_i(\boldsymbol{\theta}) - \nabla F_i(\boldsymbol{\theta}')\| \leq L_1 \|\phi_{i,0}^{\star}(\boldsymbol{\theta}) - \phi_{i,0}^{\star}(\boldsymbol{\theta}')\| + \frac{\hat{L}_2}{\lambda} \|\phi_{i,0}^{\star}(\boldsymbol{\theta}) - \phi_{i,0}^{\star}(\boldsymbol{\theta}')\| \|\nabla F_i(\boldsymbol{\theta}')\|$$
$$+ \frac{\hat{L}_1}{\lambda} \|\nabla F_i(\boldsymbol{\theta}) - \nabla F_i(\boldsymbol{\theta}')\|$$

which implies:

$$(1 - \frac{\hat{L}_1}{\lambda}) \|\nabla F_i(\boldsymbol{\theta}) - \nabla F_i(\boldsymbol{\theta}')\| \leq \left(L_1 + \frac{\hat{L}_2}{\lambda} \|\nabla F_i(\boldsymbol{\theta}')\|\right) \|\phi_{i,0}^{\star}(\boldsymbol{\theta}) - \phi_{i,0}^{\star}(\boldsymbol{\theta}')\|$$

Let's define, $\mathscr{L}(\boldsymbol{\theta}) := L_1/4 + \frac{\hat{L}_2}{4\lambda} \|\nabla F_i(\boldsymbol{\theta})\|$. Exchanging $\boldsymbol{\theta}$ and $\boldsymbol{\theta}'$, and assuming $\lambda \geq 2\hat{L}_1$, we get:

$$\|\nabla F_i(\boldsymbol{\theta}) - \nabla F_i(\boldsymbol{\theta}')\| \leq \min(\mathscr{L}(\boldsymbol{\theta}), \mathscr{L}(\boldsymbol{\theta}')) \|\phi_{i,0}^{\star}(\boldsymbol{\theta}) - \phi_{i,0}^{\star}(\boldsymbol{\theta}')\|/2 \tag{34}$$

To finish the proof, we need to bound the quantity: $\|\phi_{i,0}^{\star}(\boldsymbol{\theta}) - \phi_{i,0}^{\star}(\boldsymbol{\theta}')\|$.

We have

$$\|\phi_{i,0}^{\star}(\boldsymbol{\theta}) - \phi_{i,0}^{\star}(\boldsymbol{\theta}')\| = \|\boldsymbol{\theta} - \frac{1}{\lambda}\nabla \hat{f}_i(\phi_{i,0}^{\star}(\boldsymbol{\theta})) - \left(\boldsymbol{\theta}' - \frac{1}{\lambda}\nabla \hat{f}_i(\phi_{i,0}^{\star}(\boldsymbol{\theta}'))\right)\|$$
$$= \|\boldsymbol{\theta} - \boldsymbol{\theta}' - \frac{1}{\lambda}\left(\nabla \hat{f}_i(\phi_{i,0}^{\star}(\boldsymbol{\theta})) - \nabla \hat{f}_i(\phi_{i,0}^{\star}(\boldsymbol{\theta}'))\right)\|$$
$$\leq \|\boldsymbol{\theta} - \boldsymbol{\theta}'\| + \frac{\hat{L}_1}{\lambda} \|\phi_{i,0}^{\star}(\boldsymbol{\theta}) - \phi_{i,0}^{\star}(\boldsymbol{\theta}')\|$$

Again, choosing $\lambda \geq 2\hat{L}_1$, we get:

$$\|\phi_{i,0}^{\star}(\boldsymbol{\theta}) - \phi_{i,0}^{\star}(\boldsymbol{\theta}')\| \leq 2\|\boldsymbol{\theta} - \boldsymbol{\theta}'\|.$$

Plugging the last inequality in Eq (34), we get:

$$\|\nabla F_i(\boldsymbol{\theta}) - \nabla F_i(\boldsymbol{\theta}')\| \leq \min(\mathscr{L}(\boldsymbol{\theta}), \mathscr{L}(\boldsymbol{\theta}')) \|\boldsymbol{\theta} - \boldsymbol{\theta}'\|,$$

which finishes the proof.

## C.5 Convergence of NormalizedGD and ClippedGD in the deterministic setting

Consider a differentiable function $F$ satisfying the following property for all $\boldsymbol{\theta}, \boldsymbol{\theta}' \in \Theta$:

$$\|\nabla F(\boldsymbol{\theta}) - \nabla F(\boldsymbol{\theta}')\| \leq \min(\mathscr{L}(\boldsymbol{\theta}), \mathscr{L}(\boldsymbol{\theta}')) \|\boldsymbol{\theta} - \boldsymbol{\theta}'\|,$$

and for any function $\mathscr{L}$.

Then we have:

$$F(\boldsymbol{\theta}') = F(\boldsymbol{\theta}) + \nabla F(\boldsymbol{\theta})^{\top}(\boldsymbol{\theta}' - \boldsymbol{\theta}) + \int_0^1 \left[\nabla F(\boldsymbol{\theta} + t(\boldsymbol{\theta}' - \boldsymbol{\theta})) - \nabla F(\boldsymbol{\theta})\right]^{\top}(\boldsymbol{\theta}' - \boldsymbol{\theta})dt$$

Thus

$$|F(\boldsymbol{\theta}') - F(\boldsymbol{\theta}) - \nabla F(\boldsymbol{\theta})^{\top}(\boldsymbol{\theta}' - \boldsymbol{\theta})| \leq |\int_0^1 \left[\nabla F(\boldsymbol{\theta} + t(\boldsymbol{\theta}' - \boldsymbol{\theta})) - \nabla F(\boldsymbol{\theta})\right]^{\top}(\boldsymbol{\theta}' - \boldsymbol{\theta})dt|$$
$$\leq \int_0^1 \mathscr{L}(\boldsymbol{\theta}) t \|\boldsymbol{\theta}' - \boldsymbol{\theta}\|^2 dt$$
$$= \frac{\mathscr{L}(\boldsymbol{\theta})}{2} \|\boldsymbol{\theta}' - \boldsymbol{\theta}\|^2.$$

In particular,

$$F(\boldsymbol{\theta}') \leq F(\boldsymbol{\theta}) + \nabla F(\boldsymbol{\theta})^\top (\boldsymbol{\theta}' - \boldsymbol{\theta}) + \frac{\mathscr{L}(\boldsymbol{\theta})}{2}\|\boldsymbol{\theta}' - \boldsymbol{\theta}\|^2$$

Let's consider a general GD update: $\boldsymbol{\theta}' = \boldsymbol{\theta} - \eta \nabla F(\boldsymbol{\theta})$, where $\eta$ might depend on $\boldsymbol{\theta}$. For this update, we have

$$F(\boldsymbol{\theta}') \leq F(\boldsymbol{\theta}) - \eta\|\nabla F(\boldsymbol{\theta})\|^2 + \frac{\mathscr{L}(\boldsymbol{\theta})\eta^2}{2}\|\nabla F(\boldsymbol{\theta})\|^2$$
$$= F(\boldsymbol{\theta}) - \eta(1 - \frac{\mathscr{L}(\boldsymbol{\theta})\eta}{2})\|\nabla F(\boldsymbol{\theta})\|^2$$

It is easy to see that $\eta = \frac{1}{\mathscr{L}(\boldsymbol{\theta})}$ minimizes the right-hand side of the above inequality, which leads to:

$$F(\boldsymbol{\theta}) - F(\boldsymbol{\theta}') \geq \frac{\|\nabla F(\boldsymbol{\theta})\|^2}{2\mathscr{L}(\boldsymbol{\theta})}$$

One important observation here is that the optimal step size is the inverse of the generalized smoothness, thus if the smoothness depends on the norm of the gradient or other quantities, the optimal step size depends on them too.

Let's now discuss the special case where $\mathscr{L}(\boldsymbol{\theta}) = \mathscr{L}_0 + \mathscr{L}_1\|\nabla F(\boldsymbol{\theta})\|$.

In this case, we have:

$$F(\boldsymbol{\theta}^t) - F(\boldsymbol{\theta}^{t+1}) \geq \frac{\|\nabla F(\boldsymbol{\theta}^t)\|^2}{2(\mathscr{L}_0 + \mathscr{L}_1\|\nabla F(\boldsymbol{\theta}^t)\|)}.$$

For a given precision $\varepsilon$, the goal is to bound the number of steps $t$ necessary to reach $\|\nabla F(\boldsymbol{\theta}^t\| \leq \varepsilon$.

Before reaching the goal above, we naturally have two regimes, a first one for which $\|\nabla F(\boldsymbol{\theta}^t\| \geq \mathscr{L}_0/\mathscr{L}_1$ and another one where $\varepsilon \leq \|\nabla F(\boldsymbol{\theta}^t\| \leq \mathscr{L}_0/\mathscr{L}_1$.

If we are in the first regime, we have

$$F(\boldsymbol{\theta}^t) - F(\boldsymbol{\theta}^{t+1}) \geq \frac{(\mathscr{L}_0/\mathscr{L}_1)^2}{4\mathscr{L}_0} = \frac{\mathscr{L}_0}{4\mathscr{L}_1^2}.$$

Whereas if we were in the second regime, then we would have:

$$F(\boldsymbol{\theta}^t) - F(\boldsymbol{\theta}^{t+1}) \geq \frac{\varepsilon^2}{4\mathscr{L}_0}.$$

All in all, as long as $\|\nabla F(\boldsymbol{\theta}^t\| \geq \varepsilon$, we have

$$F(\boldsymbol{\theta}^t) - F(\boldsymbol{\theta}^{t+1}) \geq \min(\frac{\varepsilon^2}{4\mathscr{L}_0}, \frac{\mathscr{L}_0}{4\mathscr{L}_1^2})..$$

Let $K$ be the number of steps necessary to reach the first index $t$ such that $\|\nabla F(\boldsymbol{\theta}^t\| \leq \varepsilon$. Assuming that the function $F$ is lower bounded and denoting $\Delta = F(\boldsymbol{\theta}^0) - \inf F$, then

$$\Delta \geq K\min(\frac{\varepsilon^2}{4\mathscr{L}_0}, \frac{\mathscr{L}_0}{4\mathscr{L}_1^2}).$$

Thus $K \leq \frac{\Delta}{\min(\frac{\varepsilon^2}{4\mathscr{L}_0}, \frac{\mathscr{L}_0}{4\mathscr{L}_1^2})} \leq \frac{4\mathscr{L}_0\Delta}{\varepsilon^2} + \frac{4\mathscr{L}_1^2\Delta}{\mathscr{L}_0}.$

## C.6 Convergence of ClippedGD in the stochastic setting

To treat the stochastic setting, we can use Theorem 3.2 from (Zhang et al., 2020a) with $\beta = 0$ which guarantees a convergence in at most $\mathcal{O}\left(\Delta \tilde{\sigma}^2 \max(\frac{\mathcal{L}_0}{\varepsilon^4}, \frac{\mathcal{L}_1^4}{\mathcal{L}_0^3})\right)$ for $\tilde{\sigma} \neq 0$.

# D Experimental Details

## D.1 Quadratic objective experiment

We can equivalently write a quadratic objective that represents the task loss as:

$$f_i(\boldsymbol{\theta}) = \hat{f}_i(\boldsymbol{\theta}) = \frac{1}{2}\boldsymbol{\theta}^\top \boldsymbol{A}_i \boldsymbol{\theta} + \boldsymbol{\theta}^\top \boldsymbol{b}_i,$$

In this case, it is easy to show that:

$$\phi_{i,v}^\star(\boldsymbol{\theta}) = ((1+v)\boldsymbol{A}_i + \lambda \boldsymbol{I})^{-1}(\lambda \boldsymbol{\theta} - (1+v)\boldsymbol{b}_i)$$

and

$$\phi_i^\star(\boldsymbol{\theta}) = \phi_{i,0}^\star(\boldsymbol{\theta}) = (\boldsymbol{A}_i + \lambda \boldsymbol{I})^{-1}(\lambda \boldsymbol{\theta} - \boldsymbol{b}_i)$$

The meta-gradient of task $i$ has the expression:

$$\nabla F_i(\boldsymbol{\theta}) = \lambda (\boldsymbol{A}_i + \lambda \boldsymbol{I})^{-1} \nabla_\phi \mathcal{L}_i(\phi_i^\star(\boldsymbol{\theta}))$$
$$= \lambda (\boldsymbol{A}_i + \lambda \boldsymbol{I})^{-1}\left(\boldsymbol{A}_i(\boldsymbol{A}_i + \lambda \boldsymbol{I})^{-1}(\lambda \boldsymbol{\theta} - \boldsymbol{b}_i) + \boldsymbol{b}_i\right)$$

And it is not difficult to verify that indeed $\nabla F_i(\boldsymbol{\theta}) = -\lambda \frac{d\phi_i^\star(\boldsymbol{\theta})}{dv}\Big|_{v=0}$.

Because we have the exact expression of the meta-gradient, we can compute it exactly and compare the relative precision of different approximation methods. This is what we did in Figure 1.

## D.2 MNIST-1D Classification Setup

The MNIST-1D dataset is a 1D version of MNIST consisting of 12,000 examples of 40-dimensional vectors across 10 classes. The meta-learning task is constructed as follows:

- **Task Structure**: We use a 3-way 5-shot configuration with 10 query samples per class.

- **Model Architecture**: A Functional Conv1D model is used, consisting of three convolutional layers (16 filters, kernel size 3, padding 1) each followed by BatchNorm, ReLU, and MaxPool1D. The final layer is a linear map to the 3-way output.

- **Hyperparameters**: We chose the best hyperparameters on a logarithmic grid from 1 to $1e-5$ for both the inner learning rate and the meta-learning rate with the Adam optimizer. The inner loop consists of 5 steps.

- **Algorithm Constants**: For FO-B-MAML and iMAML, we set $\lambda = 2.0$. FO-B-MAML uses a perturbation scale $v = 0.1$, and iMAML uses 5 Conjugate Gradient steps.

## D.3 CNN Memory Scalability Benchmarking

The memory scalability experiments were conducted using an activation-heavy ConvNet architecture. The peak memory usage was measured using `torch.cuda.max_memory_allocated` for GPU execution.

- **Procedure**: We varied the "Number of Channels" from 16 to 256.

- **Sequential Execution**: To maintain the memory footprint of Mem($\theta$), FO-B-MAML solves the perturbed subproblems $\tilde{\phi}_{i,v}$ and $\tilde{\phi}_{i,-v}$ sequentially, discarding intermediate activations and storing only the resulting parameter estimates.

- **Baseline Comparison**: MAML memory usage includes the storage of the full computation graph for all inner steps to allow for backpropagation, leading to the observed $O(\text{Inner Steps} \times \text{Activations})$ scaling.

## D.4   Omniglot Few-Shot Classification Setup

We evaluate FO-B-MAML on the Omniglot dataset to assess performance on high-dimensional, multi-class image data.

- **Task Structure**: We follow the standard 5-way 1-shot and 5-way 5-shot protocols. Each task includes 15 query samples per class for meta-gradient calculation.

- **Model Architecture**: We utilize the standard four-layer convolutional backbone common in MAML literature (64 filters per layer, $3 \times 3$ convolutions, ReLU activations, and $2 \times 2$ max-pooling).

- **Optimization**: We employ 5 adaptation steps in the inner loop with $\lambda = 2.0$. The outer loop uses the Adam optimizer with a meta-learning rate selected via grid search.

- **Parameters**: The perturbation scale is set to $v = 0.1$, consistent with our MNIST-1D experiments.

## D.5   Transformer Memory Scalability Benchmarking

To investigate the "activation bottleneck" in attention-based models, we conduct memory profiling on a Transformer architecture.

- **Architecture**: We use a standard Transformer block consisting of multi-head self-attention (8 heads) and a feed-forward network (512 hidden units).

- **Scaling Protocol**:

  - **Model Width**: We vary the embedding dimension ($d_{model}$) from 64 to 512 while keeping the sequence length fixed at 128.
  - **Sequence Length**: We vary the input sequence length ($L$) from 32 to 512 while keeping $d_{model} = 128$ to measure the impact of quadratic attention memory.

- **Memory Measurement**: Peak memory is recorded as the maximum GPU memory allocated during a full meta-gradient update. FO-B-MAML utilizes sequential execution for the symmetric subproblems to ensure the memory footprint remains independent of the activation maps of the inner trajectory.

