# OpenReview forum: "A New First-Order Meta-Learning Algorithm with Convergence Guarantees"
_TMLR — Accepted by TMLR_

### Review · Reviewer_qJgk · 2026-03-29

**Summary Of Contributions:**

The paper proposes FO-B-MAML,a  first-order meta-learning algorithm derived from a bi-level optimization perspective. The key contribution is a new formulation of the meta-gradient as the derivative of the solution to a perturbed inner problem, enabling efficient estimation via finite-difference schemes without requiring second-order derivatives or backpropagation through the inner loop. Compared to existing first-order methods , the proposed approach offers controllable and theoretically reduced bias, along with rigorous convergence guarantees to stationary points. The method is also memory-efficient, as it avoids storing inner-loop activations, making it scalable to deep models. Additionally, the paper provides theoretical insight into the non-standard smoothness properties of the MAML objective, motivating the use of gradient clipping or normalization.

**Audience:**

Yes

**Audience Explanation:**

This work would primarily interest meta-learning and optimization researchers focused on scaling algorithms like MAML to deep, activation-heavy architectures where memory overhead is a primary bottleneck. Specifically, those working on bi-level optimization and few-shot learning would value the new first-order gradient estimator that provides theoretical convergence guarantees and a superior bias rate compared to existing methods like Reptile or FO-MAML. Furthermore, researchers in optimization theory would find the justification for clipped-gradient method through the characterization of the objective’s smoothness constant particularly relevant for stabilizing training in non-standard settings.

**Claims And Evidence:**

Yes

**Claims Explanation:**

The paper provides evidence for FO-B-MAML by proving it converges to a stationary point with an improved bias rate, strictly enhancing previous first-order theory. Empirically, it achieves high accuracy on MNIST-1D (~0.95), tracking closely with second-order MAML while significantly outperforming other first-order methods like FO-MAML. Furthermore, it demonstrates superior scalability by bypassing the "activation bottleneck," maintaining a flat memory footprint even when scaling to deep, activation-heavy CNNs with up to 250 channels.

**Requested Changes:**

Though the paper shows improvement on MNIST, but it will be better to add other such as ImageNet data to cover the impact of improvement on larger dataset. In terms of arch, the paper shows the improvement on CNN arch only, how does it compare when working on transformer?  The success of the symmetric estimator depends on the choice of the perturbation scale and regularization  but the paper provides limited guidance on how to automatically tune these for new tasks. So adding more on how someone can efficiently tune this parameter will help as well or any auto-tuning methods.

---

> ### Author Response · Authors · 2026-04-26
>
> **Summary of Changes:**
> We thank Reviewer qJgk for the positive assessment of our "activation bottleneck" analysis and theoretical justification for clipped-gradient methods. We have implemented the following changes based on your recommendations:
>
> * **Transformer Scalability:** We conducted new scalability experiments in Section 6.4 and Appendix B using a Transformer backbone. As shown in Figures 3 and 6, FO-B-MAML maintains a significantly flatter memory footprint as embedding dimensions and sequence lengths increase.
> * **Practical Tuning Guidance:** We have added a dedicated "Empirical Analysis of Hyperparameter Sensitivity" in Appendix A.
>     * **Appendix A.1** introduces a **"Step-Aware" selection rule** for the perturbation scale $\nu$: for coarse adaptation (low $K$), a larger $\nu \in [0.1, 0.5]$ suppresses numerical noise, while for high-precision adaptation (high $K$), $\nu$ can be decreased to $0.01$ to minimize the bias.
>     * **Appendix A.2** analyzes the regularization strength $\lambda$, illustrating the trade-off between adaptation distance and numerical stability in Figure 5.
> * **Auto-Tuning Discussion:** We added a discussion in Section 7 and Appendix A.2 regarding how $\lambda$ can be treated as a meta-parameter. Our framework allows estimating $\nabla_{\lambda}F$ using a similar finite-difference identity, potentially enabling the model to learn optimal "adaptation stiffness" automatically.

---

### Review · Reviewer_EA28 · 2026-03-30

**Summary Of Contributions:**

The paper proposes FO-B-MAML, a first-order meta-learning algorithm derived from a bi-level formulation of MAML, with the aim of reducing second-order computational and memory costs while retaining provable convergence and competitive empirical performance.  The paper reformulates MAML as a purely bi-level optimization problem in which the task-adapted parameter is defined as the solution to a regularized inner optimization problem, rather than as the output of a particular optimization trajectory. This removes path dependence and motivates a new expression for the meta-gradient: instead of differentiating through the full inner-loop dynamics, the meta-gradient is characterized as the derivative of the solution to a perturbed inner problem with respect to a scalar perturbation parameter. Based on this identity, the authors introduce FO-B-MAML, which estimates the meta-gradient using finite differences between solutions of perturbed inner problems. Two estimators are studied: a forward-difference estimator and a symmetric-difference estimator. The paper derives bias bounds for both, arguing that the symmetric estimator has a lower bias than the forward estimator when the inner problem is solved approximately. The theoretical analysis also examines the smoothness of the meta-objective, claiming that standard global Lipschitz smoothness does not generally hold and that a generalized smoothness condition better fits the problem, motivating normalized or clipped gradient methods as outer optimizers. FO-B-MAML is reported to produce progressively more accurate meta-gradient estimates as inner optimization improves, to approach second-order MAML performance on MNIST-1D while clearly outperforming FO-MAML, and to maintain a much flatter memory footprint than MAML on deep CNNs, especially in activation-heavy architectures.

**Audience:**

Yes

**Audience Explanation:**

Meta-Learning is an important contemporary machine learning problem of importance across the TMLR audience.

**Broader Impact Concerns:**

I don't have any BI concerns, but it would be helpful if the authors could add a brief discussion of the implications of this work in the General Discussion section.

**Claims And Evidence:**

Yes

**Claims Explanation:**

First, the paper has a clear and meaningful methodological idea: recasting the meta-gradient as the derivative of a perturbed inner solution is elegant and conceptually distinct from simply dropping higher-order terms as in FO-MAML. That makes the contribution more than an engineering tweak. Second, the work tries to connect algorithm design, convergence theory, and systems considerations. The paper not only proposes an estimator, but also analyzes estimator bias, discusses generalized smoothness, and links the theory to clipped or normalized gradient methods; that breadth is valuable. The empirical section addresses both optimization quality and practical scalability. The synthetic experiment is useful because it provides a setting with a known meta-gradient, and the CNN memory experiment targets a real, practical pain point of second-order meta-learning methods. The memory claim is also visually clear in Figures 2 and 3.

That said, the theory rests on strong assumptions, especially strong convexity of the inner objective and substantial smoothness requirements on train/test losses. Those assumptions are hard to reconcile with standard deep meta-learning settings, so the gap between theory and the neural-network experiments is significant. Also, the empirical evaluation is narrow. The main benchmark is MNIST-1D, and the paper does not test on more standard few-shot benchmarks such as Omniglot, miniImageNet, tieredImageNet, or meta-learning reinforcement-learning settings.

**Requested Changes:**

In addition to addressing the weaknesses I mentioned above, I would suggest including further information about other related works.
The paper does discuss the main optimization-based meta-learning baselines: MAML, FO-MAML, Reptile, iMAML, and ES-MAML, and it positions itself primarily against them. That coverage is appropriate for the core argument. However, the related-work discussion could be stronger in two ways. First, it could engage more deeply with the broader bi-level optimization and hypergradient approximation literature, rather than just meta-learning papers. The core contribution is fundamentally a hypergradient estimation technique for a regularized inner problem, so stronger positioning relative to implicit differentiation, truncated backpropagation, Neumann-series approximations, and finite-difference hypergradient methods would help. Second, the paper could discuss more carefully whether its idea is specific to MAML-style few-shot learning or part of a larger family of bilevel methods for hyperparameter optimization and nested learning. That would clarify novelty and scope.

---

> ### Author Response · Authors · 2026-04-26
>
> **Summary of Changes:**
> We appreciate Reviewer EA28’s recognition of our perturbation-based meta-gradient identity as "elegant" and conceptually distinct. We have addressed your concerns regarding literature positioning and the theoretical assumptions as follows:
>
> * **Expanded Literature Positioning:** Per your suggestion, we have significantly revised Section 2 (Related Work) to engage more deeply with the broader bi-level optimization (BLO) and hypergradient approximation literature. We now explicitly discuss our method's relationship to implicit differentiation and finite-difference hypergradient methods, highlighting how FO-B-MAML avoids the Jacobian inversions required by iMAML.
> * **Bridging Theory and Practice:** To address the gap between standard deep learning settings and our strong convexity assumption, we added a "Discussion of Assumptions" in Section 5.1 and a stability analysis in Appendix A.2. We provide empirical evidence in Figure 5 showing that the proximal regularization $\lambda$ directly improves the condition number of the inner-optimization matrix, effectively "convexifying" the local landscape to stabilize the solution path even in non-convex settings.
> * **Scope and Generalization:** The General Discussion (Section 7) explains that while we focus on MAML-style few-shot learning, the identity in Proposition 1 is general and can be applied to other nested learning problems, such as Hyperparameter Optimization (HPO). We also think the Finite Difference approximation of the meta-gradient can be applied more generally to bi-level optimization problems and will lead to improved rates (under stronger regularity assumptions).
> * **Broader Impact:** Following your suggestion, we have included a brief discussion on the practical implications of memory-efficient meta-learning in the General Discussion (Section 7).

---

### Review · Reviewer_aDGP · 2026-04-19

**Summary Of Contributions:**

This paper proposes FO-B-MAML, a memory-efficient variant of Model-Agnostic Meta-Learning based on bi-level optimization. The key idea is to approximate the meta-gradient using only first-order information, thereby avoiding the computational and memory overhead associated with higher-order derivatives. To achieve this, the authors introduce a novel approximation scheme based on perturbing the inner optimization problem, which enables efficient estimation of the meta-gradient. Building on this idea, the paper presents a MAML-based framework with improved memory efficiency.

Strengths

* The paper addresses an important challenge in meta-learning, namely the computational and memory complexity of MAML-style methods.
* The proposed approach is conceptually simple and avoids higher-order gradient computation, which is appealing from both theoretical and practical perspectives.

Weaknesses

* Limited positioning within existing literature:
    There is a substantial body of work on MAML variants and first-order meta-learning methods, including both empirical and theoretical studies. The paper provides insufficient discussion and comparison with these prior works, making it difficult to clearly understand its novelty and relative contribution.
* Overly simplistic experimental setup:
    The experimental validation is limited to a very simple benchmark (MNIST-1D, 3-way 5-shot). This is not sufficient to demonstrate the general effectiveness of the proposed method. Evaluation on more diverse and realistic meta-learning benchmarks is necessary to support the claims.
* Unclear theoretical contribution:
    While the paper presents theoretical analysis, the core contribution is not clearly articulated. The assumptions used in the analysis are numerous, but their practical relevance is not sufficiently discussed.

**Audience:**

Yes

**Audience Explanation:**

Modifying MAML toward improved memory and computational efficiency, while maintaining or enhancing performance, is likely to provide valuable inspiration for researchers in this area.

**Claims And Evidence:**

No

**Claims Explanation:**

Although the theoretical analysis supports the claims, it relies on numerous assumptions that are insufficiently discussed, and the experimental validation remains limited.

**Requested Changes:**

* Provide a more comprehensive and critical discussion of prior work on MAML variants and first-order meta-learning methods.
* Clearly state what is novel in the theoretical analysis
* The assumptions used in the theoretical development should be carefully discussed
*  The current experiments are too limited. The paper should include more diverse benchmarks like Omniglot,

---

> ### Author Response · Authors · 2026-04-26
>
> **Summary of Changes:**
> We thank Reviewer aDGP for the constructive feedback regarding the positioning and experimental breadth of our work. We have implemented the following major revisions:
>
> * **Comprehensive Positioning:** We significantly expanded Section 2 (Related Work) to position FO-B-MAML within the broader Bi-Level Optimization (BLO) and hypergradient approximation literature. We now explicitly compare our approach to implicit differentiation (iMAML) and Neumann-series approximations, highlighting how our estimators avoid the Jacobian inversions and memory-heavy trajectory tracking required by those methods.
> * **Experimental Expansion (Omniglot):** To address the concern regarding simplistic benchmarks, we added results on the **Omniglot (5-way 1-shot and 5-shot)** benchmark in Section 6.3. As shown in Table 2, FO-B-MAML achieves competitive accuracy ($99.24 \pm 0.29$) compared to established baselines
> * **Clarification of Theoretical Novelty & Assumptions:** We revised our contributions and articulated our core gradient identity (Proposition 1) more clearly. We added a "Discussion of Assumptions" in Section 5.1, clarifying that Assumption 4 (Strong Convexity) is a standard requirement for well-defined bi-level problems to ensure a unique inner solution mapping $\theta \mapsto \phi^*$. In practice, the proximal regularization $\lambda$ effectively "convexifies" the local landscape, a behavior we analyze empirically in Appendix A.2.
> * **Additional Experiments:** We added new experiments in Appendix A measuring the sensitivity of the perturbation scale $\nu$ (Figure 4) and regularization strength $\lambda$ (Figure 5). Furthermore, we included new benchmarks on memory scaling for Transformer architectures in Section 6.4 (Figure 3) and Appendix B (Figure 6) to demonstrate the algorithm's scalability to modern, activation-heavy models.

---

### Author Response · Authors · 2026-05-21

Dear Reviewers,

Since we haven't heard from you, we are writing to inquire whether you have had a chance to review the revisions we made to the paper. Per your demand, we added more experiments (Few-shot learning on the Omniglot dataset and memory scaling for Transformer architectures), expanded the related work section, and added clarifications throughout the paper (concerning assumptions and how they have been used in previous works, like IMAML, and also extension of the ideas).

We are here if you need any clarification or if you have any more questions about the paper. Otherwise, we ask the AC to communicate the final decision.

Please let us know what needs to be done next.

Best,
The authors.

---

### Decision · Action_Editor_GaSz · 2026-06-18

**Recommendation:** Accept as is

**Audience:**

Yes

**Audience Explanation:**

The paper should be of interest to researchers working on meta-learning, bilevel optimization, hypergradient estimation, and memory-efficient training algorithms.

**Claims And Evidence:**

Yes

**Claims Explanation:**

The submission makes a clear methodological contribution: it proposes FO-B-MAML, a first-order bilevel formulation of MAML in which the meta-gradient is estimated through finite differences of perturbed inner solutions, thereby avoiding explicit second-order differentiation and backpropagation through the full inner optimization trajectory. The technical claims are supported by theoretical analysis of the proposed estimators, including bias bounds and convergence guarantees, and the empirical claims are supported by experiments comparing optimization behavior, accuracy, and memory scaling against relevant MAML-style baselines. The reviewers generally agreed that the central idea is technically meaningful and that the revision improved the paper by expanding related work, adding Omniglot and Transformer memory-scaling experiments, and providing additional discussion of perturbation and regularization choices.
At the same time, the evidence is not without limitations. In particular, the assumptions underlying the theory remain strong relative to the nonconvex deep-learning settings that motivate the work, and the empirical validation remains somewhat limited compared with the full range of modern meta-learning benchmarks. Nevertheless, I judge that the paper's main claims are appropriately qualified and sufficiently supported for TMLR acceptance, especially given that the primary contribution is algorithmic and theoretical rather than a claim of broad empirical dominance.